# TEST TIME ADAPTATION WITH AUXILIARY TASKS

## ABSTRACT

This work tackles a key challenge in Test Time Adaptation (TTA): adapting on limited data. This challenge arises naturally from two scenarios. **(i)** Current TTA methods are limited by the bandwidth with which the stream reveals data, since conducting several adaptation steps on each revealed batch from the stream will lead to overfitting. **(ii)** In many realistic scenarios, the stream reveals insufficient data for the model to fully adapt to a given distribution shift. We tackle the first scenario problem with auxiliary tasks where we leverage unlabeled data from the training distribution. In particular, we propose distilling the predictions of an originally pretrained model on clean data during adaptation. We found that our proposed auxiliary task significantly accelerates the adaptation to distribution shifts. We report a performance improvement over the state of the art by 1.5% and 6% on average across all corruptions on ImageNet-C under episodic and continual evaluation, respectively. To combat the second scenario of limited data, we analyze the effectiveness of combining federated adaptation with our proposed auxiliary task across different models even when different clients observe different distribution shifts. We find that not only federated averaging enhances adaptation, but combining it with our auxiliary task provides a notable 6% performance improvement over previous TTA methods.

## 1 INTRODUCTION

Deep Neural Networks (DNNs) have achieved remarkable success in providing state-of-the-art results in several applications (Ranftl et al., 2021; He et al., 2016; Deng et al., 2009). However, their performance severely deteriorates whenever a shift exists between training and testing distributions (Hendrycks et al., 2021a;b). Such distribution shifts are likely to occur in real-world settings in the form of changes in weather conditions (Hendrycks & Dietterich, 2019), camera parameters (Kar et al., 2022), data compression or, in extreme cases, adversarial perturbations (Goodfellow et al., 2015). Needless to say, mitigating or adapting to the effects of such distribution shifts is crucial to the safe deployment of DNNs in many use cases, *e.g.*, self-driving cars.

Test Time Adaptation (TTA) (Sun et al., 2020; Liu et al., 2021) attempts to resolve this problem by closing the gap between the model performance when tested with or without distribution shifts. In particular, TTA adapts a pretrained model at test time by optimizing a proxy objective function on a stream of *unlabeled* data in an online fashion (Wang et al., 2021). The recent progress in TTA showed great success in improving performance under distribution shifts in several scenarios (Niu et al., 2022; Wang et al., 2022; Yuan et al., 2023). However, and to prevent overfitting, all TTA methods in the literature conduct a single adaptation step on each received batch at test time (Niu et al., 2023; Nguyen et al., 2023). This limits the efficacy of TTA methods by the bandwidth of the stream and the amount of data that the model receives, hampering their online performance. Furthermore, the current paradigm of TTA focuses on updating a single model at a time, assuming the stream will reveal enough data to capture the underlying distribution shift. Nonetheless, in many realistic settings, the stream of data accessible to an individual model might be too scarce to enable adequate adaptation. In such scenarios, we might accelerate adaptation by leveraging other models being adapted to similar domain shifts in a collaborative and federated fashion (Jiang & Lin, 2023).

In this work, we tackle the aforementioned lack of data in Test Time Adaptation by proposing an auxiliary task that can be optimized during test time. Since the amount of data from a given distribution shift is limited by the bandwidth of the stream, we follow Kang et al. (2023); Gao et al. (2022); Niu et al. (2022) in leveraging unlabeled data from the training distribution. We first show

how one could simply employ the same proxy objective of previous TTA methods on unlabeled clean data to accelerate the adaptation to distribution shifts. Based on this observation, we propose DISTA (Distillation-based TTA) a better auxiliary objective that distills the predictions of the originally pretrained model on clean unlabeled data during adaptation. We assess the effectiveness of DISTA on two different benchmarks and 3 different evaluation protocols where consistent and significant performance improvements are attained. In summary, our contributions are threefold:

1. We present a methodology to analyze the effectiveness of auxiliary tasks on accelerating the adaptation under distribution shift through lookahead analysis. We show that one can leverage clean *unlabeled* data to better adapt to distribution shifts.

2. We propose DISTA; a TTA method with a distillation based auxiliary task. We conduct comprehensive experimental analysis on the two standard and large-scale TTA benchmarks ImageNet-C (Hendrycks & Dietterich, 2019) and ImageNet-3DCC (Kar et al., 2022) where we show how DISTA improves the performance over state-of-the-art methods by a significant margin (1.5% under episodic evaluation and 6-8% under continual evaluation).

3. We further analyze a novel and realistic scenario where each individual model is presented with insufficient amount of data for adaptation. We first show how federated learning facilitates adaptation in this case, even when the observed distribution shift varies among clients. Further, we show how DISTA provides a large performance gain (6% on ImageNet-C) over state-of-the-art methods in this federated setup.

## 2 METHODOLOGY

**Preliminaries** Test Time Adaptation (TTA) studies the practical problem of adapting pretrained models to *unlabeled* streams of data from an unknown distribution that potentially differs from the training one. In particular, let $f_\theta : \mathcal{X} \to \mathcal{P}(\mathcal{Y})$ be a classifier parametrized by $\theta$ that maps a given input $x \in \mathcal{X}$ to a probability simplex over $k$ labels (*i.e.* $f_\theta^i(x) \geq 0, \|f_\theta(x)\|_1 = 1$). During the training phase, $f_\theta$ is trained on some *source* data $\mathcal{D}_s \subseteq \mathcal{X} \times \mathcal{Y}$, but at test time, it is presented with a stream of data $\mathcal{S}$ that might be differently distributed from $\mathcal{D}_s$. In this work, we focus on covariate shifts, i.e. changes in the distribution over the input space $\mathcal{X}$ due to, for instance, visual corruptions caused by changes in weather conditions faced by self-driving systems. TTA defines a learner $g(\theta, x)$ that adapts the network parameters $\theta$ and/or the received unlabeled input $x$ at test time to enhance the performance of the model under such distribution shifts. Throughout, we use distribution and domain shift interchangeably. Formally, and following the online learning notation (Shalev-Shwartz, 2011; Cai et al., 2021; Ghunaim et al., 2023; Alfarra et al., 2023), we describe the interaction at a time step $t \in \{0, 1, \ldots, \infty\}$ between a TTA method $g$ and the stream of unlabeled data $\mathcal{S}$ as:

1. $\mathcal{S}$ reveals a sample $x_t$.

2. $g$ adapts $x_t$ to $\hat{x}_t$, $\theta_t$ to $\hat{\theta}_t$, generates a prediction $\hat{y}_t$, and updates parameters via $\theta_{t+1} = \alpha\theta_t + (1 - \alpha)\hat{\theta}_t$ with $0 \leq \alpha \leq 1$.

The main paradigm in TTA employs an unsupervised objective function to be optimized on-the-fly at test time to circumvent performance drops caused by domain shift. Wang et al. (2021) observed a strong correlation between the entropy of the output prediction for a given batch of inputs and the error rate. Based on that, Wang et al. (2021) proposed to minimize the entropy of the output prediction for a given batch of inputs at test time through:

$$\theta_{t+1} = \arg\min_\theta \mathbb{E}_{x_t \sim \mathcal{S}} \left[ E\left(f_\theta(x_t)\right) \right] \qquad \text{with} \quad E\left(f_\theta(x_t)\right) = -\sum_i f_\theta^i(x_t) \log f_\theta^i(x_t). \quad (1)$$

Note that the optimization problem is usually solved with a single gradient descent step to avoid overfitting network parameters on each received batch. It is noteworthy that this approach has demonstrated a great success only when the received batches **(i)** have diverse sets of labels and **(ii)** relate to a single type of domain shift (Niu et al., 2023). In previous work, Niu et al. (2022) attempted to circumvent these drawbacks by deploying a data selection procedure, while Yuan et al. (2023) leveraged a balanced episodic memory that have inputs with a diverse set of labels.

## 2.1 TEST TIME ADAPTATION WITH AUXILIARY TASKS

TTA imposes many challenges due to its realistic setup. One main challenge is that the learner needs to adapt the model to *unlabeled* data revealed from the stream in an online manner. Another

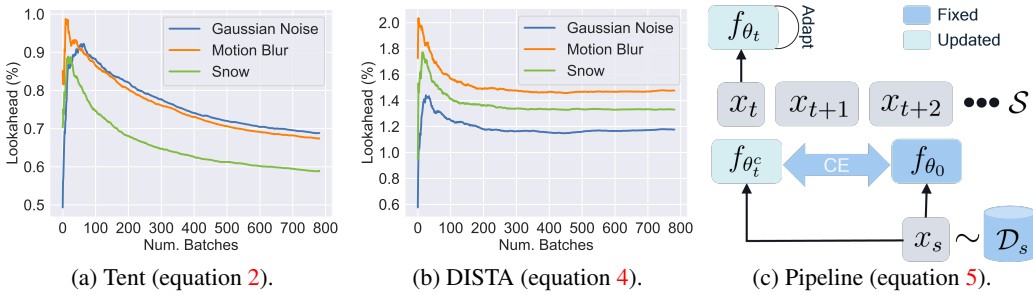

(a) Tent (equation 2).         (b) DISTA (equation 4).         (c) Pipeline (equation 5).

Figure 1: **Lookahead Analysis and Pipeline.** (a) Running mean of lookahead over observed batches when employing Tent on both data revealed from the stream and $\mathcal{D}_s$. (b) Running mean of lookahead over observed batches using DISTA. (c) Pipeline summarizing our proposed DISTA.

important challenge we identify is that the learner is constrained by the speed of the stream and the amount of revealed data. That is, the model needs to adapt on a limited amount of data. The faster the learner adapts to the distribution shift, the better its online performance. However, most TTA methods in the literature conduct a single adaptation step to prevent overfitting model parameters to each received batch. That is, even when new batches are revealed slowly enough to allow multiple optimization steps, the learner $g$ cannot benefit from this additional time. This naturally begs the question: can we enhance the adaptation speed of TTA methods in such setting? In this work, we address this question through the lens of auxiliary tasks (Lyle et al., 2021).

Auxiliary tasks (Liebel & Körner, 2018) are additional loss terms that indirectly optimize the desired objective function. A simple auxiliary loss function to be optimized is the TTA method, *e.g.* minimizing entropy on $x_t$ for one more step, which results in overfitting on the revealed batch from the stream. We take a step back and ask the following question: what could an adaptation model access at step $t$ other than $x_t$? EATA (Niu et al., 2022), for instance, leveraged $\mathcal{D}_s$ for calculating the anti-forgetting regularizer while DDA (Gao et al., 2022) used $\mathcal{D}_s$ to train a diffusion model to project $x_t$ into the source domain. More recently, Kang et al. (2023) condensed $\mathcal{D}_s$ to construct a set of labeled examples per class used for adaptation. While one could potentially access labeled samples $\mathcal{D}_s$ for the aforementioned approach, several applications do not allow accessing this labeled distribution (*e.g.* training procedure can be outsourced with private training data). Note that, however, one could get unlabeled data from this distribution cheaply. For example, one could store few *unlabeled* data examples at clear weather conditions (for autonomous driving applications) as a proxy for source distributions before deploying the model in an episodic memory, following Yuan et al. (2023). Having said that, a natural question could arise: how can we use unlabeled samples from $\mathcal{D}_s$ to better adapt on distribution shifts in $\mathcal{S}$?

We first examine a simple auxiliary task: during test time, adapt the model not only on the data revealed from the stream (*i.e.* $x_t$), but also on a sample $x_s \sim \mathcal{D}_s$. For example, let $g(\theta, x)$ be the entropy minimization approach in equation 1. One could solve the following objective function:

$$\min_{\theta} \; \mathbb{E}_{x_t \sim \mathcal{S}} E\left(f_\theta(x_t)\right) \; + \; \mathbb{E}_{x_s \sim \mathcal{D}_s} E\left(f_\theta(x_s)\right). \tag{2}$$

At first glance, it is unclear whether the additional term in the loss function would effectively facilitate adaptation to domain shifts in $\mathcal{S}$. Thus, to better analyze the effect of the auxiliary term, we solve the optimization problem in equation 2 with the following alternative optimization approach:

$$\theta_t^c = \theta_t - \gamma \nabla_\theta \left[E\left(f_\theta(x_t)\right)\right] \qquad \theta_{t+1} = \theta_t^c - \gamma \nabla_\theta \left[E\left(f_\theta(x_s)\right)\right]. \tag{3}$$

Note that the gradients in the first and second SGD steps are evaluated at $\theta_t$ and $\theta_t^c$, respectively. Now, we can study the effect of our simple auxiliary task by measuring the improvement on the entropy after optimizing the auxiliary task; denoted as *lookahead* (Fifty et al., 2021), and defined as

$$\text{Lookahead}(\%) = 100 \times \left(1 - \frac{E\left(f_{\theta_{t+1}}(x_t)\right)}{E\left(f_{\theta_t^c}(x_t)\right)}\right).$$

Note, the higher the lookahead, the better the auxiliary task is at minimizing the desired objective.

We conduct experiments on ImageNet-C benchmark (Hendrycks & Dietterich, 2019) where we fix $f_{\theta_0}$ to be ResNet-50 (He et al., 2016) pretrained on ImageNet dataset (Deng et al., 2009). We

measure the lookahead over samples revealed from the stream for when $\mathcal{S}$ contains one of 3 domain shifts (Gaussian Noise, Motion Blur, and Snow) and we take $\mathcal{D}_s$ as a subset of unlabeled images from the training set. For each received batch $x_t$ from the stream $\mathcal{S}$, we sample a batch $x_s$ from $\mathcal{D}_s$ with the same size for simplicity. Figure 1a summarizes the results. We can observe that the simple auxiliary task of minimizing the entropy of predictions on source data has, surprisingly, a positive impact on the desired task (*i.e.* minimizing entropy on corrupted data). This hints that one could accelerate the convergence of adaptation on corrupted data by leveraging unsupervised auxiliary tasks on source data. We highlight that through our lookahead analysis, one could analyze the effectiveness of different auxiliary tasks in TTA. Next, we describe our proposed auxiliary task.

## 2.2 DISTA: Distillation Based Test Time Adaptation

In Section 2.1, we analyzed the positive impact of one example of auxiliary task, observing that entropy minimization on source data does improve adaptation to domain shifts. Next, we propose a better and more powerful auxiliary task. We distill a saved copy of the original pretrained model $f_{\theta_0}$ during adaptation on samples from the source distribution. More precisely, we replace the entropy minimization term on the source data with a cross-entropy loss between the predictions of $f_{\theta_t}$ and $f_{\theta_0}$. We also employ a data selection scheme whereby we update the model on samples with low entropy, following Niu et al. (2022). Our overall objective function can be described as follows:

$$\min_{\theta} \mathbb{E}_{x_t \sim \mathcal{S}} \lambda_t(x_t) E\left(f_\theta(x_t)\right) \;+\; \mathbb{E}_{x_s \sim \mathcal{D}_s} \lambda_s(x_s) CE\left(f_\theta(x_s), f_{\theta_0}(x_s)\right) \tag{4}$$

$$\text{where } \lambda_t(x) = \frac{\mathbb{1}_{\{E(f_{\theta_t}(x)) < E_0\}} \cdot \mathbb{1}_{\{\cos(f_{\theta_t}(x), m^{t-1}) < \epsilon\}}}{\exp(E(f_{\theta_t}(x)) - E_0)}, \; \lambda_s(x) = \frac{\mathbb{1}_{\{E(f_{\theta_t}(x)) < E_0\}}}{\exp(E(f_{\theta_t}(x)) - E_0)}$$

where $\mathbb{1}_{\{.\}}$ is an indicator function that takes the value 1 if the condition $\{.\}$ is satisfied and 0 otherwise, $\epsilon$ and $E_0$ are positive thresholds, and $m^{t-1}$ is the moving average of the prediction vector. We note here that both $\lambda_t$ and $\lambda_s$ are data selection functions that prevent updating the model on unreliable or redundant samples. To assess the effectiveness of our proposed auxiliary task, we follow our setup in Section 2.1 and consider the following alternating optimization approach:

$$\theta_t^c = \theta_t - \gamma \nabla_\theta \left[\lambda_t(x_t) E\left(f_\theta(x_t)\right)\right] \qquad \theta_{t+1} = \theta_t^c - \gamma \nabla_\theta \left[\lambda_s(x_s) CE\left(f_\theta(x_s), f_{\theta_0}(x_s)\right)\right]. \tag{5}$$

Hence, we can now measure the lookahead and analyze how effective our approach is for adaptation. We replicate our setup from Section 2.1 and report the results in Figure 1b. We find that our proposed auxiliary task has a positive lookahead over all observed batches. It is worth mentioning that we observe similar results with all types of domain shifts we considered, as indicated by more detailed lookahead results that we defer to appendix for the sake of conciseness. That is, solving our auxiliary task on clean data in an online fashion helps the model to adapt faster and better to distribution shifts presented in the stream $\mathcal{S}$. Please refer to Figure 1c for an illustration of DISTA.

**Intuition behind DISTA.** First, based on our observation in Section 2.1, minimizing the entropy of the predictions on clean data can accelerate adaptation and hence improve online performance. However, besides the clean data, we also have access to the pretrained model $f_{\theta_0}$. Therefore, we can combine both sources of information to obtain the richer auxiliary task of knowledge distillation (Hinton et al., 2015), which can improve performance in similar settings (Hong et al., 2021). Further, our auxiliary task in DISTA allows adapting a pretrained model to domain shifts while being close to $f_{\theta_0}$ in the output space. We hypothesize this allows for a more stable adaptation that prevents $f_{\theta_t}$ from diverging and overfitting on each presented domain shift by $\mathcal{S}$. We argue that this approach is richer than the simple entropy minimization in equation 2 while being more beneficial and flexible than regularizing the parameter space as in EATA (Niu et al., 2022).

## 3 Related Work

**Test Time Training (TTT)** aims at updating a pretrained model at test time on the received unlabeled data when there is a distribution shift between training and testing data (Sun et al., 2020). This is usually done by including a self-supervised loss function during the training process (*e.g.* predicting the rotation angle (Gidaris et al., 2018)) that will be later used at test time (Liu et al., 2021; Chen et al., 2022; Tzeng et al., 2017). It is noteworthy that such approaches, while being effective in mitigating performance drops under distribution shifts, are less practical as they require control over the training process, and thus are not readily applicable to any pretrained model (Sun et al., 2020).

Table 1: **Episodic Evaluation on ImageNet-C Benchmark with ResNet-50.** We report the error rate (lower is better) for each corruption. We adapt the model to each corruption independently in episodic evaluation. DISTA improves over the previous state-of-the-art EATA on all domain shifts.

| | Noise | | | Blur | | | | Weather | | | | Digital | | | | |
| | Gauss | Shot | Impul | Defoc | Glass | Motion | Zoom | Snow | Frost | Fog | Bright | Contr | Elastic | Pixel | Jpeg | Avg. |
|---|---|---|---|---|---|---|---|---|---|---|---|---|---|---|---|---|
| Source | 97.8 | 97.1 | 98.1 | 82.1 | 90.2 | 85.2 | 77.5 | 83.1 | 76.7 | 75.6 | 41.1 | 94.6 | 83.0 | 79.4 | 68.4 | 82.0 |
| AdaBN | 84.9 | 84.3 | 84.3 | 85.0 | 84.7 | 73.6 | 61.1 | 65.8 | 66.9 | 52.1 | 34.8 | 83.3 | 56.1 | 51.1 | 60.3 | 68.5 |
| BN | 84.6 | 83.9 | 83.8 | 80.1 | 80.2 | 71.7 | 60.4 | 65.4 | 65.2 | 51.6 | 34.6 | 76.3 | 54.4 | 49.7 | 59.2 | 66.7 |
| SHOT | 73.1 | 69.8 | 72.0 | 76.9 | 75.9 | 58.5 | 52.7 | 53.3 | 62.2 | 43.8 | 34.6 | 82.6 | 46.0 | 42.3 | 48.9 | 59.5 |
| TTAC | 71.3 | 70.3 | 70.8 | 82.1 | 77.4 | 63.9 | 53.9 | 49.9 | 55.5 | 43.9 | 32.8 | 81.4 | 43.7 | 41.1 | 46.7 | 59.0 |
| Tent | 70.3 | 68.2 | 69.0 | 72.2 | 73.0 | 58.8 | 50.7 | 52.7 | 59.0 | 42.7 | 32.7 | 72.9 | 45.6 | 41.4 | 47.6 | 57.1 |
| SAR | 69.5 | 69.7 | 69.0 | 71.2 | 71.7 | 58.1 | 50.5 | 52.9 | 57.9 | 42.7 | 32.7 | 62.9 | 45.5 | 41.6 | 47.8 | 56.2 |
| EATA | 64.0 | 62.1 | 62.5 | 66.9 | 66.9 | 52.5 | 47.4 | 48.2 | 54.2 | 40.2 | 32.2 | 54.6 | 42.2 | 39.2 | 44.7 | 51.9 |
| DISTA | **62.2** | **59.9** | **60.6** | **65.3** | **65.3** | **50.4** | **46.2** | **46.6** | **53.1** | **38.7** | **31.7** | **53.2** | **40.8** | **38.1** | **43.5** | **50.4** |

Table 2: **Episodic Evaluation on ImageNet-3DCC Benchmark with ResNet-50.** We compare our proposed DISTA with the previous state-of-the-art EATA in terms of error rate (lower is better).

| | Bit Error | Quant. | Far Focus | Flash | Fog | H256 ABR | H256 CRF | Noise | Low Light | Near Focus | XY Blur | Z Blur | Avg. |
|---|---|---|---|---|---|---|---|---|---|---|---|---|---|
| EATA | 91.5 | 58.9 | 47.8 | 71.0 | 62.2 | 72.4 | 67.3 | 56.1 | 46.8 | 38.6 | 64.9 | 52.7 | 60.9 |
| DISTA | **91.4** | **57.9** | **47.0** | **70.2** | **61.8** | **71.5** | **66.3** | **54.1** | **45.5** | **38.0** | **63.8** | **51.5** | **59.9** |

**Test Time Adaptation** (TTA) relaxes the assumption of altering the training process and solely optimizes a given pretrained model at test time (Liang et al., 2020; Boudiaf et al., 2022; Su et al., 2022). Earlier approaches showed a strong impact of adapting the statistics of the normalization layers on reducing the error rate under distribution shifts (Li et al., 2016; Schneider et al., 2020; Mirza et al., 2022). This was followed by the seminal work of Wang et al. (2021) which showed a correlation between the entropy of predicted samples and the error rate. This observation initiated a line of work that minimizes the entropy of the predictions at test time such as TENT (Wang et al., 2021), MEMO (Zhang et al., 2021b), and the more powerful EATA (Niu et al., 2022) and SAR (Niu et al., 2023). Later approaches employed data augmentations at test time to enhance invariance to distribution shifts (Nguyen et al., 2023; Yuan et al., 2023). More closely to our work, some TTA methods distilled the training data for adaptation through model optimization (Kang et al., 2023), feature matching (Mirza et al., 2023), or input projection via diffusion models (Gao et al., 2022). In this work, we approach TTA through the lens of auxiliary tasks. In essence, we show how one could leverage unlabeled data samples from the training distribution to accelerate the adaptation.

**Evaluation Protocols in TTA.** The predominant evaluation protocol in TTA is the episodic evaluation: adapting the pretrained model to one type of distribution shift at a time (*e.g.* fog) where the environment reveal batches of data with mixed categories. More recently, a line of work tackled more challenging setups such as continual evaluation (Wang et al., 2022), practical evaluation (Yuan et al., 2023), a computationally budgeted evaluation (Alfarra et al., 2023), and federated evaluation (Jiang & Lin, 2023). In this work, we experiment with our proposed DISTA under different evaluation protocols showing its superiority to previous methods in the literature in different cases.

## 4 EXPERIMENTS

**Setup.** We follow prior art in focusing our experiments on the image classification task (Niu et al., 2023; Su et al., 2022; Liang et al., 2020) where $f_\theta$ is a model pretrained on ImageNet (Deng et al., 2009). In our experiments, we consider different architectures including the standard ResNet-50 (He et al., 2016), the smaller ResNet-18, ResNet-50-GN (replacing Batch Normalization Layers with Group Normalization layers), and Vision Transformers (ViT) (Ranftl et al., 2021), following Niu et al. (2023). Regarding the evaluation benchmarks, we consider two large scale standard benchmarks in the TTA literature; ImageNet-C (Hendrycks & Dietterich, 2019) and the more realistic ImageNet-3DCC (Kar et al., 2022). We fix the severity level in our experiments to 5 and evaluate on all corruptions presented in both of the aforementioned datasets. Unless stated otherwise and following prior work (Wang et al., 2021; Niu et al., 2022; Wang et al., 2022), we report results for ResNet-50 He et al. (2016) as the architecture $f_\theta$ and assume that the stream $\mathcal{S}$ reveals batches of data with a size of 64. Nonetheless, Section 4.4.2 presents results under different architectures and batch sizes. Please refer to the appendix for further experimental details.

Table 3: **Continual Evaluation on ImageNet-C with ResNet-50.** We report the average error rate per (lower is better) corruption when $\mathcal{S}$ contains a sequence of domain shifts (ordered from left to right) followed by the clean validation set of ImageNet. DISTA improves over previous state-of-the-art by 6% on average across all corruptions and on clean data.

| | Noise | | | Blur | | | | Weather | | | | Digital | | | | | |
|---|---|---|---|---|---|---|---|---|---|---|---|---|---|---|---|---|---|
| | Gauss | Shot | Impul | Defoc | Glass | Motion | Zoom | Snow | Frost | Fog | Bright | Contr | Elastic | Pixel | Jpeg | Avg. | Val. |
| CoTTA | 77.2 | 66.9 | 63.1 | 75.1 | 71.5 | 69.4 | 67.1 | 71.9 | 71.2 | 67.1 | 62.0 | 73.1 | 69.1 | 66.1 | 68.0 | 69.3 | 61.4 |
| SAR | 68.6 | 61.7 | 61.8 | 72.6 | 69.8 | 65.1 | 57.6 | 63.7 | 64.1 | 52.8 | 41.2 | 67.6 | 52.8 | 49.4 | 52.5 | 60.1 | 34.1 |
| EATA | 64.0 | 58.8 | 59.2 | 69.2 | 68.1 | 62.8 | 56.4 | 58.5 | 60.6 | 48.4 | 39.2 | 58.9 | 49.0 | 45.4 | 48.7 | 56.5 | 32.7 |
| DISTA | **62.4** | **56.9** | **57.0** | **63.5** | **62.9** | **51.4** | **46.3** | **48.1** | **53.5** | **40.1** | **32.8** | **52.8** | **42.5** | **38.9** | **43.3** | **50.2** | **26.3** |

Table 4: **Continual Evaluation on ImageNet-3DCC with ResNet-50.** We compare DISTA to the previous state-of-the-art EATA in terms of average error rate per corruption when $\mathcal{S}$ contains a sequence of domain shifts (ordered from left to right) followed by the clean ImageNet validation set. DISTA improves over EATA by $\sim 8\%$ on average across all corruptions and by 9% on clean data.

| | Bit Error | Quant. | Far Focus | Flash | Fog | H256 ABR | H256 CRF | Noise | Low Light | Near Focus | XY Blur | Z Blur | Avg. | Val. |
|---|---|---|---|---|---|---|---|---|---|---|---|---|---|---|
| EATA | 91.5 | 71.5 | 57.2 | 74.6 | 66.6 | 79.0 | 75.0 | 66.9 | 55.9 | 48.5 | 70.6 | 59.3 | 68.1 | 35.8 |
| DISTA | **91.0** | **61.2** | **48.9** | **70.4** | **61.4** | **72.1** | **66.0** | **55.1** | **45.1** | **39.2** | **63.4** | **50.8** | **60.4** | **26.5** |

In our experiments, we consider a total of 8 TTA baselines from the literature. In particular, we analyze methods that adapt the statistics of BN layers, such as Adabn (Li et al., 2016) and BN (Schneider et al., 2020); the clustering approach TTAC-NQ (Su et al., 2022); SHOT (Liang et al., 2020), which maximizes the mutual information; the continual adaptation method CoTTA (Wang et al., 2022); entropy minimization approaches, such as Tent (Wang et al., 2021); and the state-of-the-art methods that employ data point selection procedures, like SAR (Niu et al., 2023) and EATA (Niu et al., 2022). We follow the official implementation of all baselines with their recommended hyperparameters.

Regarding our proposed DISTA, for each received $x_t$, we sample $x_s$ from $\mathcal{D}_s$ with an equivalent batch size. We employ our alternating optimization approach described in equation 5. We fix $\mathcal{D}_s$ to be a randomly selected subset of ImageNet training set[1]. We consider different approaches to solve our proposed auxiliary objective function with more analysis in Section 4.4.1. For the evaluation protocols, we consider the standard and simplest episodic evaluation in Section 4.1, the more challenging life-long continual evaluation in Section 4.2, and a novel federated evaluation in Section 4.3. Finally, we present ablation studies and analysis in Section 4.4.

## 4.1 EPISODIC EVALUATION

We start with the simple episodic evaluation, following the common practice in the TTA literature (Liang et al., 2020; Wang et al., 2021; Niu et al., 2022). In this setting, the stream $\mathcal{S}$ contains data from a single type of domain shift w.r.t to the training distribution (*e.g.* fog). We report the error rates for all 15 corruptions in the ImageNet-C benchmark in Table 1 for different TTA methods.

We observe that **(i)** DISTA sets new state-of-the-art results in the episodic evaluation by outperforming EATA. In particular, we found that our auxiliary distillation task reduces the error rate under all corruptions by a significant 1.5% on average and by 2% on shot noise and motion blur. Table 2 shows that similar improvements are presented on the more realistic and challenging ImageNet-3DCC benchmark. This result demonstrates the effectiveness of DISTA in accelerating the convergence of entropy minimization on data received from the stream, as evidenced in Figure 1b. That is, the faster the model is in adapting to earlier batches, the better the performance of the model on later batches revealed by the stream.

## 4.2 CONTINUAL EVALUATION

Next, we consider the more realistic and challenging continual evaluation protocol. In this setting, the stream $\mathcal{S}$ presents the learner with a sequence of domain shifts. We follow Kang et al. (2023) in constructing the stream $\mathcal{S}$ by concatenating all corruptions in the ImageNet-C benchmark. We report the results on different domain orders in the appendix due to space limitations. Further, and to assess the performance of the model on the original source distribution upon adaptation, we follow Alfarra

---

[1]We experimented with $\mathcal{D}_s$ being a subset of the validation set and did not notice any changes in our results.

Table 5: **Federated Evaluation on ImageNet-C with ResNet-50.** We split the data belonging to each corruption into 50 clients (no overlap) and report the average error rate per corruption. We consider the local training (-L) where there is no communication across clients and the federated adaptation (-F) when clients with the same domain shift category communicate their models for averaging. For example, clients with Noise corruption (Gaussian, Shot, and Impulse) average their models every communication round. We observe that federated adaptation reduces the error rate over local adaptation. Further, DISTA improves over other methods in both scenarios.

| | Noise | | | Blur | | | | Weather | | | | Digital | | | | |
|---|---|---|---|---|---|---|---|---|---|---|---|---|---|---|---|---|
| | Gauss | Shot | Impul | Defoc | Glass | Motion | Zoom | Snow | Frost | Fog | Bright | Contr | Elastic | Pixel | Jpeg | Avg. |
| Tent-L | 83.6 | 82.9 | 82.9 | 84.7 | 84.6 | 73.3 | 60.7 | 65.4 | 66.6 | 51.8 | 34.9 | 82.7 | 55.8 | 50.5 | 59.6 | 68.0 |
| Tent-F | 72.6 | 69.7 | 69.6 | 75.3 | 74.8 | 65.8 | 57.0 | 57.8 | 61.1 | 47.1 | 36.5 | 73.4 | 50.4 | 46.8 | 51.4 | 60.6 |
| EATA-L | 82.3 | 81.4 | 81.8 | 83.8 | 83.6 | 72.2 | 59.8 | 64.0 | 65.7 | 50.5 | 34.3 | 81.1 | 54.7 | 49.5 | 57.9 | 66.8 |
| EATA-F | 68.8 | 66.0 | 66.0 | 72.5 | 72.5 | 64.6 | 59.0 | 54.5 | 59.1 | 45.6 | 38.2 | 64.0 | 49.7 | 45.8 | 49.7 | 58.4 |
| DISTA-L | 81.1 | 79.6 | 80.4 | 82.7 | 82.6 | 70.4 | 58.2 | 62.2 | 64.3 | 48.7 | 34.2 | 79.4 | 53.3 | 48.1 | 55.8 | 65.4 |
| DISTA-F | 62.8 | 58.8 | 58.9 | 66.8 | 66.0 | 54.6 | 48.2 | 50.5 | 54.9 | 40.6 | 33.8 | 56.6 | 44.8 | 40.2 | 44.6 | 52.1 |

et al. (2023) by appending the clean validation set of ImageNet as a last domain in the stream $\mathcal{S}$. For this evaluation setup, we consider three strong continual adaptation methods: CoTTA (Wang et al., 2022), SAR (Niu et al., 2023), and EATA (Niu et al., 2022) that are designed for life-long adaptation. Table 3 summarizes the results on ImageNet-C where the order of domains presented to the learner follows the order in the table (from left to right). We accompany the reported error rate on each corruption with the average error rate under all domain shifts. We further adapt the model on the clean validation set (source distribution) at the end of the stream (last column).

We observe that **(ii)** DISTA sets a new state-of-the-art in continual evaluation by outperforming EATA by a notable 6% on average across all corruptions. It is worth noting that the performance gap is particularly wide for snow, motion and zoom blur, where DISTA reduces the error rate by 10% or more. **(iii)** Furthermore, while all considered methods suffer from a significant performance drop on the source distribution, our distillation auxiliary task prevents forgetting the source domain and reduces the error rate on clean validation data by more than 6%, recovering the performance of the non-adapted model. This goes to show that, while our auxiliary task enhances the convergence speed of adaptation, this improved convergence does not come at the cost of overfitting to each adapted domain. In fact, our distillation loss helped in better life-long adaptability, and importantly, not forgetting the original source domain. Notably, the performance of DISTA under continual evaluation was not substantially different from that under episodic evaluation. This demonstrates the stability that our auxiliary task provides in the adaptation process. We also complement our experiments with continual evaluation on ImageNet-3DCC dataset and report the results in Table 4. We observe similar results on this more challenging dataset where we outperform the previous state of the art, EATA, by 8% on average across all corruptions and by 9% on the clean validation set.

## 4.3 FEDERATED EVALUATION

**Motivation.** In all of the previous evaluation schemes, we focused on adapting a single model having access to the entire stream of data $\mathcal{S}$. However, in many realistic scenarios there might be several deployed models, and the data received by each one of them individually might not be enough for adaptation. Federated learning (Konečnỳ et al., 2016; Zhang et al., 2021a) shines in this setting by allowing different models to communicate their updates privately with a server that aggregates the information and sends back a more powerful global model. The aggregation step is usually done through federated averaging (Konečnỳ et al., 2016), where the global model is the average of the weights of the local models. In this section we analyze a novel federated evaluation setup of TTA.

**Setup.** We consider a category-wise federated TTA setup where clients (*i.e.* models) adapting to the same category of domain shifts (*e.g.* all weather corruptions in ImageNet-C) communicate their updates for a better global adaptation. We divide the data belonging to a single domain into $N$ non-overlapping subsets where each client adapts to a stream of data coming from one of these subsets. Further, we allow all clients to have $M$ communication rounds with the server that aggregates the updates and sends back the global model. We consider the full participation setup where all clients participate in each adaptation and communication round. For instance, in the weather conditions case, all clients adapting to snow, frost, fog, and brightness will communicate their models to be aggregated via federated averaging. Note that setting $N = 1$ and $M = 0$ recovers the episodic

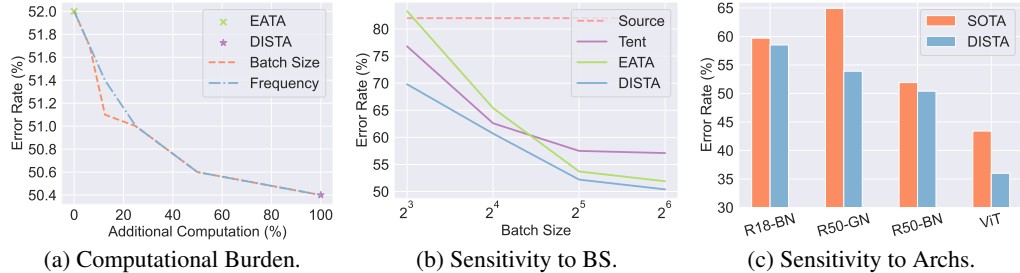

(a) Computational Burden.  (b) Sensitivity to BS.  (c) Sensitivity to Archs.

Figure 2: **Analysis on DISTA.** (a) Shows the tradeoff between the performance improvement that DISTA provides compared to EATA *vs* the additional computational requirement. (b) Shows the robustness of the performance gain of DISTA under different batch sizes. (c) Shows consistent performance gains of DISTA under different architectures when compared to EATA (Niu et al., 2022) (ResNet-18, 50) and SAR (Niu et al., 2023) (ResNet 50-GN, ViT).

evaluation in Section 4.1. In our experiments, we set $N = 50$ and compare the performance of local adaptation (*i.e.* setting $M = 0$) and the federated adaptation with $M = 4$ which results in a communication round each 4 adaptation steps.

**Results.** We report the error rates on the 4 corruption categories in Table 5 for Tent, EATA, and DISTA where (-L) represents local adaptation and (-F) represents federated adaptation. We observe that **(v)** Conducting federated adaptation provides consistently lower error rates than adapting each client solely on their own local stream of data. This result is consistent for all considered methods. Note that the performance gain is despite the fact that in each communication round, models adapting to different domain shifts are being aggregated. **(vi)** Furthermore, DISTA is consistently outperforming all other baselines under both the local and federated adaptation setups. Specifically, DISTA improves over EATA by a notable 6% on average in the federated adaptation setup.

## 4.4 ANALYSIS

### 4.4.1 COMPUTATIONAL AND MEMORY BURDEN

In the previous experimental results, we demonstrated the effectiveness of our proposed DISTA in different evaluation schemes and benchmarks. Now, we delve into fine-grained analysis of our proposed auxiliary task. We first observe that the second update step in equation 5 has a similar cost to the adaptation step on $x_t$ as we sample $x_s$ with the same size. This makes the overall cost of DISTA $2\times$ the cost of updating using ETA. Next, we discuss some tricks to accelerate DISTA.

**Parallel updates.** The main bottleneck in the update step in equation 5 is that $\theta_{t+1}$ is a function of $\theta_t^c$. That is, the two optimization steps on $x_t$ and $x_s$ are done sequentially. Assuming access to sufficient compute memory, and inspired by federated averaging, one could alternatively solve the DISTA optimization problem in equation 4 with

$$\theta_t^c = \theta_t - \gamma \nabla_\theta \left[\lambda_t(x_t) E\left(f_\theta(x_t)\right)\right] \qquad \theta_t^s = \theta_t - \gamma \nabla_\theta \left[\lambda_s(x_s) CE\left(f_\theta(x_s), f_{\theta_0}(x_s)\right)\right] \qquad (6)$$

and set $\theta_{t+1} = (\theta_t^c + \theta_t^s)/2$. This will allow both update steps on $x_t$ and $x_s$ to be conducted in parallel, minimizing the latency of DISTA. We found that this approach, with the very same hyperparameters, yields similar results to the solver in equation 5. Further details are left for the appendix.

**Memory restrictive setup.** While the parallel approach in equation 6 reduces the latency of DISTA, it incurs larger memory costs than EATA. Thus, we consider the memory conservative case with a sequential update for DISTA. We analyze the sensitivity of DISTA to **(1)** varying the batch size of $x_s$ and **(2)** the frequency of updates on $x_s$ under a fixed batch size of 64. Figure 2a reports the average error rate on ImageNet-C under episodic evaluation. We analyze the performance of DISTA under different additional computational burdens. Note that for $0\%$ additional computation, the performance of DISTA restores the current state-of-the-art EATA. Interestingly, we observe a smooth trade-off between additional computation and performance gains. For example, with $50\%$ additional computation (*i.e.* optimizing the auxiliary task on every other batch) DISTA outperforms EATA by 1.4% on average. That is, one could save $50\%$ of the additional computation of DISTA with a very marginal drop in performance gains. We leave the rest of the discussion to the appendix.

Table 6: **Orthogonality of Auxiliary Tasks with ResNet-50.** We quantify the performance improvement of employing two different auxiliary tasks. We follow our formulation in equation 2 for Aux-Tent, and similarly for Aux-SHOT. In both scenarios, auxiliary tasks assisted the adaptation.

| | Noise | | | Blur | | | | Weather | | | | Digital | | | |
| | Gauss | Shot | Impul | Defoc | Glass | Motion | Zoom | Snow | Frost | Fog | Bright | Contr | Elastic | Pixel | Jpeg | Avg. |
|---|---|---|---|---|---|---|---|---|---|---|---|---|---|---|---|---|
| Tent | 70.3 | 68.2 | 69.0 | 72.2 | 73.0 | 58.8 | 50.7 | 52.7 | 59.0 | 42.7 | 32.7 | 72.9 | 45.6 | 41.4 | 47.6 | 57.1 |
| Aux-Tent | 68.5 | 66.4 | 66.6 | 71.1 | 71.9 | 55.8 | 49.3 | 50.8 | 60.4 | 41.6 | 32.7 | 80.8 | 44.2 | 40.5 | 46.3 | 56.5 |
| SHOT | 73.1 | 69.8 | 72.0 | 76.9 | 75.9 | 58.5 | 52.7 | 53.3 | 62.2 | 43.8 | 34.6 | 82.6 | 46.0 | 42.3 | 48.9 | 59.5 |
| Aux-SHOT | 67.1 | 64.9 | 65.7 | 69.0 | 69.9 | 55.5 | 49.8 | 50.7 | 58.7 | 42.3 | 33.3 | 68.2 | 44.4 | 41.1 | 46.5 | 55.1 |

### 4.4.2 ABLATION STUDIES

**Sensitivity to batch size.** For completeness, we analyze the sensitivity of DISTA when the stream $\mathcal{S}$ reveals batches of different sizes. In particular, we consider batch sizes in $\{64, 32, 16, 8\}$. We conduct episodic evaluation on ImageNet-C and report the average error rate on all corruptions in Figure 2b. We compare our DISTA with the non-adapted model (Source), Tent, and EATA. We observe that DISTA provides consistent performance improvement under all considered batch sizes. In fact, at batch size 8, DISTA improves upon EATA by more than 15%. It is worth noting that the data selection process of EATA hinders its effectiveness for small batch sizes, allowing Tent to outperform it, but our proposed auxiliary task seems to mitigate the same effect for DISTA.

**Experiments with different architectures.** Finally, we follow the recent work of Niu et al. (2023) and explore the effectiveness of integrating DISTA into different architectures. In particular, we consider the smaller and more efficient ResNet18, ResNet50-GN, and ViT (Ranftl et al., 2021). For all architectures, we follow Niu et al. (2023) in adapting only the normalization layers and compare the performance against EATA on ResNet18 and ResNet50, and against SAR on ResNet50-GN and ViT (best performing method). We report the results in Figure 2c where we follow our episodic evaluation on ImageNet-C. We find that DISTA consistently outperforms other baselines irrespective of the choice of the architecture. That is, our proposed distillation auxiliary loss is reducing the error rate on the four considered architectures. In fact, we found that DISTA improves over SAR under the ViT architecture by an impressive 7%, setting new state-of-the-art results. Due to limited space, we leave experiments with ViT architecture under batch size 1 along with ablating the effect of varying the size of $\mathcal{D}_s$ on DISTA's performance to the appendix. Further, we analyze the effectiveness of a variant of DISTA that leverages labeled examples from the source distribution in the appendix.

### 4.4.3 ORTHOGONALITY OF AUXILIARY TASKS

In Section 2.1, we analyzed a simple auxiliary task integrated with Tent (Wang et al., 2021) and showed its positive impact with the lookahead analysis. In this section, we analyze experimentally the benefits of auxiliary tasks on different TTA methods. In a similar spirit to our analysis in Section 2.1, we add an auxiliary task, as in equation 2, to two TTA methods; namely SHOT (Liang et al., 2020) and Tent (Wang et al., 2021). That is, for each TTA method, we conduct two adaptation steps: one on $x_t$ and one on $x_s$. For Tent, we precisely conduct the alternating optimization scheme in equation 3 while for SHOT, we replace the entropy with a mutual information term on both $x_t$ and $x_s$. Table 6 reports results for episodic evaluation on ImageNet-C.

We observe that our auxiliary task approach is orthogonal to the adaptation strategy. Both Aux-Tent and Aux-SHOT outperform their original baselines by a significant margin. For example, optimizing the auxiliary task yields a 3% error rate reduction on motion blur for both baselines. It is worth mentioning that we record a more notable performance improvement when employing the auxiliary task on SHOT (4% performance improvement on average) compared with Tent (0.6% improvement on average). We leave a more detailed analysis with more experiments to the appendix.

## 5 CONCLUSIONS

In this work, we analyzed the effectiveness of auxiliary tasks in accelerating the adaptation to distribution shifts through lookahead analysis. In particular, we showcased two scenarios for when test time adaptation suffer from limited available data for adaptation (slow stream and limited data per client). In both scenarios, our proposed DISTA provided significant performance gains. Further, we showed how DISTA is robust to the choice of architecture, batch size, and the evaluation protocol.

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

## A  Test Time Adaptation with Auxiliary Tasks

### A.1  DISTA: Distillation Based Test Time Adaptation

In Section 2.2, we showed how our proposed auxiliary task in DISTA had a positive lookahead for three corruptions from the ImageNet-C benchmark. Here, for the sake of completeness, we provide the lookahead plots for the remaining corruptions in ImageNet-C in Figure 3. We observe, similarly to our earlier findings in Section 2.2, that our auxiliary task has a consistent positive lookahead across all corruptions. That is, our distillation loss on clean data helps to better adapt to domain shifts. Note that this is already demonstrated through our extensive experimental evaluation in Sections 4.1-4.3 where DISTA consistently outperformed previous state-of-the-art TTA methods.

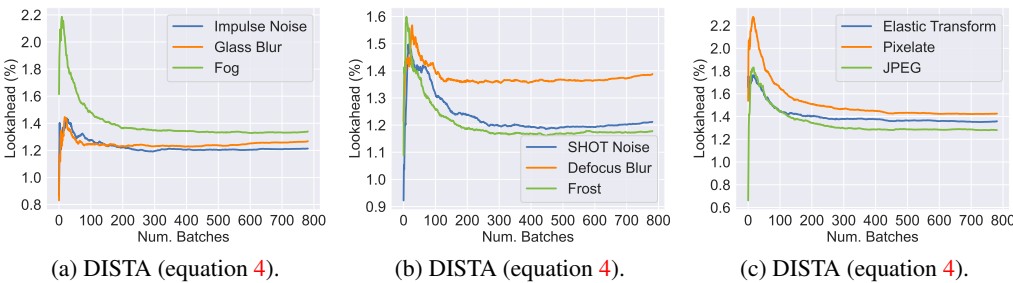

|                (a) DISTA (equation 4).                |                (b) DISTA (equation 4).                |                (c) DISTA (equation 4).                |

Figure 3: **Lookahead Analysis.** We plot the lookahead of DISTA for the 12 different corruptions from the ImageNet-C benchmark. We find that our proposed auxiliary task always yields a positive lookahead across all considered corruptions. These results corroborate our hypothesis that optimizing our distillation task on clean data helps adapting to distribution shifts.

## B  Additional Experiments

### B.1  Experimental Setup and Hyper-parameter Choices

In Section 4, we outlined our experimental setup in terms of architectures and evaluation protocols. In this section, we delve more deeply into implementation and experimental details that, due to space constraints, were not able to elaborate on in the main paper. For all baselines, we used the official code released by the authors to reproduce their results with their recommended hyperparameters. Note that all analyzed TTA methods (except SHOT) operate solely on the normalization layers of a given network. That is, $\theta$ always refers to the learnable parameters of the normalization layers (*e.g.* BatchNorm layers). Further, and following Wang et al. (2021) and Niu et al. (2022), we use an SGD optimizer with learning rate of $25 \times 10^{-4}$ and momentum of 0.9. For DISTA, we follow Niu et al. (2022) in setting $\epsilon = 5 \times 10^{-2}$ in equation 4 but pick a higher value for $E_0$; we set $E_0 = 0.5 \log(1000)$ instead of $0.4 \log(1000)$, since we observed better lookahead with modest increases in $E_0$. Yet, as we as show in a later section, we still observe better results with DISTA than with EATA even when keeping $E_0 = 0.4 \log(1000)$. Regarding Aux-Tent, we set the learning rate to $5 \times 10^{-4}$. For Aux-SHOT, the learning rate is set to the default value recommended by SHOT.

Table 7: **Continual Evaluation on ImageNet-C Under Different Domain Orders with ResNet-50.** We report the average error rate on corrupted (across all 15 corruptions) and clean domains with different random orders of domains. The first two columns are the summary of the evaluation in Section 4.2. We observe a more stable adaptation with DISTA in comparison to EATA under different domain orders where the performance gap surpasses 10%. Lower is better.

| Seed | Ordered | | 42 | | 4242 | | 424242 | | Avg. | |
|------|---------|-------|-------|-------|-------|-------|--------|-------|-------|-------|
|      | Corr.   | Clean | Corr. | Clean | Corr. | Clean | Corr.  | Clean | Corr. | Clean |
| EATA  | 56.5 | 32.7 | 63.6 | 38.5 | 64.7 | 39.4 | 65.8 | 40.4 | 62.7 | 37.8 |
| DISTA | **50.2** | **26.3** | **52.3** | **27.6** | **52.2** | **27.7** | **52.6** | **28.2** | **51.8** | **27.4** |

Table 8: **Episodic Evaluation on ImageNet-C Benchmark with ResNet-50.** We report the results of employing parallel update (DISTA-P) compared with sequential update (DISTA) to improve efficiency. We observe that both solvers yield comparable results that are consistently better than EATA. Hence, under sifficient memory availability, one can improve latency with the parallel update.

| | Noise | | | Blur | | | | Weather | | | | Digital | | | | |
| | Gauss | Shot | Impul | Defoc | Glass | Motion | Zoom | Snow | Frost | Fog | Bright | Contr | Elastic | Pixel | Jpeg | Avg. |
|---|---|---|---|---|---|---|---|---|---|---|---|---|---|---|---|---|
| EATA | 64.0 | 62.1 | 62.5 | 66.9 | 66.9 | 52.5 | 47.4 | 48.2 | 54.2 | 40.2 | 32.2 | 54.6 | 42.2 | 39.2 | 44.7 | 51.9 |
| DISTA | 62.2 | 59.9 | 60.6 | 65.3 | 65.3 | 50.4 | 46.2 | 46.6 | 53.1 | 38.7 | 31.7 | 53.2 | 40.8 | 38.1 | 43.5 | 50.4 |
| DISTA-P | 62.4 | 60.1 | 61.0 | 65.0 | 65.0 | 50.6 | 46.4 | 46.8 | 53.2 | 39.0 | 31.9 | 53.4 | 41.1 | 38.3 | 43.7 | 50.5 |

### B.2 Continual Evaluation

In Section 4.2, we evaluated DISTA under the continual learning setup where the stream $\mathcal{S}$ contains multiple distribution shifts presented one at a time. We followed the evaluation setup from Niu et al. (2022) regarding the order of types of domain shift in the stream $\mathcal{S}$. Here, and for completeness, we evaluate DISTA and compare it to EATA when the order of different domains is shuffled. We report the results across 3 random seeds that control the randomness of domains in $\mathcal{S}$ in Table 7.

We observe that while randomly shuffling the domains of ImageNet-C in the stream $\mathcal{S}$ has a large impact on the performance of EATA, DISTA is much more robust against such variation. That is, we report a performance drop of 7-9% for EATA when the corruptions are randomly ordered, and thus more severe shifts between presented domains are expected compared to a nicely ordered sequence. However, the same effect is virtually absent when using DISTA, for which the added randomness in domain order had little effect on the performance either on corrupted or clean domains. This brings another demonstration of the stability of DISTA under different evaluation schemes.

### B.3 Analysis

#### B.3.1 Computational Burden

In Section 4.4.1, we discussed an alternative approach of solving the DISTA optimization problem for the sake of improving efficiency. In particular, we considered a parallel update in equation 3. We compare the performance of the alternating solver (DISTA) and parallel solver (DISTA-P) against EATA in Table 8 (Episodic evaluation on ImageNet-C). We observe the performance of DISTA-P is on par with that of DISTA, with both variants outperforming EATA by a significant margin. That is, our proposed auxiliary task is boosting the performance irrespective of the deployed solver. Hence, one can improve the efficiency (latency) by employing the parallel solver for our proposed objective in equation 4 when sufficient memory is available.

#### B.3.2 Ablation Studies

In Section 4.4.2, we analyzed the sensitivity of DISTA under different batch sizes when compared against Tent and EATA. We showed how DISTA is much more stable than both approaches when tested with very small batch sizes. Here, we step up the game and analyze DISTA under the smallest batch size of 1 where most TTA methods fail.

SAR (Niu et al., 2023) provided state-of-the-art results under this realistic evaluation (batch size of 1) by employing a stable update and leveraging a ViT architecture, where Layer Normalization layers are independent of the batch size. In that regard, we fix the architecture in this section to ViT

Table 9: **Episodic Evaluation on ImageNet-C Under Batch Size of 1 with ViT.** We compare DISTA and SAR under batch size of 1 when emplying the ViT architecture. We observe that DISTA significantly outperforms SAR under this setting.

| | Noise | | | Blur | | | | Weather | | | | Digital | | | | |
| | Gauss | Shot | Impul | Defoc | Glass | Motion | Zoom | Snow | Frost | Fog | Bright | Contr | Elastic | Pixel | Jpeg | Avg. |
|---|---|---|---|---|---|---|---|---|---|---|---|---|---|---|---|---|
| SAR | 54.2 | 56.4 | 53.4 | 46.4 | 49.2 | 42.5 | 46.9 | 41.3 | 46.7 | 31.1 | 23.8 | 34.3 | 41.8 | 31.1 | 33.7 | 42.19 |
| DISTA | **47.5** | **48.7** | **46.6** | **44.8** | **44.7** | **40.2** | **42.0** | **32.9** | **34.2** | **27.4** | **22.0** | **32.4** | **35.8** | **29.7** | **32.6** | **37.43** |

Table 10: **Episodic Evalutation on ImageNet-C of SHOT with different auxiliary components with ResNet-50.** We experiment with auxiliary components when combined with SHOT. (Aux.) represents applying SHOT on both clean and corrupted data. (Fil) adds filtering unreliable examples. (DIS) replaces SHOT as an auxiliary task with our distillation task.

| | Noise | | | Blur | | | | Weather | | | | Digital | | | | |
|---|---|---|---|---|---|---|---|---|---|---|---|---|---|---|---|---|
| | Gauss | Shot | Impul | Defoc | Glass | Motion | Zoom | Snow | Frost | Fog | Bright | Contr | Elastic | Pixel | Jpeg | Avg. |
| SHOT | 73.1 | 69.8 | 72.0 | 76.9 | 75.9 | 58.5 | 52.7 | 53.3 | 62.2 | 43.8 | 34.6 | 82.6 | 46.0 | 42.3 | 48.9 | 59.5 |
| + Aux | 67.1 | 64.9 | 65.7 | 69.0 | 69.9 | 55.5 | 49.8 | 50.7 | 58.7 | 42.3 | 33.3 | 68.2 | 44.4 | 41.1 | 46.5 | 55.1 |
| + Fil. | 66.2 | 64.1 | 64.3 | 68.5 | 68.7 | 54.9 | 49.0 | 50.0 | 56.7 | 41.7 | 32.7 | 64.2 | 44.0 | 40.6 | 45.9 | 54.1 |
| + DIS | **64.9** | **62.6** | **62.7** | **67.1** | **66.9** | **52.9** | **47.9** | **48.6** | **55.4** | **40.5** | **32.4** | **61.8** | **42.9** | **39.3** | **44.7** | **52.7** |

where we update the learnable parameters of the normalization layers. We compare the performance of SAR and DISTA under this setting and with batch size of 1 in Table 9. We observe that DISTA significantly outperforms SAR under this setup. In particular, DISTA provides an average of $\sim$ 5% reduction on the error rate under episodic evaluation on ImageNet-C. This performance gain is consistent across all corruptions in the ImageNet-C benchmark.

### B.3.3 ORTHOGONALITY OF AUXILIARY TASKS

In Section 4.4.3, we showed how our auxiliary task approach is orthogonal to the underlying TTA method. In particular, we showed in Table 6 how applying an auxiliary task on clean data helps with either a Tent-like or a SHOT-like approach. Here we delve more onto this orthogonality. For the sake of this study, we pick SHOT as a TTA method. We report in Table 10 the effect of different auxiliary components on the overall performance of SHOT. Note that we fix the architecture to ResNet-50 and conduct episodic evaluation on the ImageNet-C benchmark.

First, we observe that employing an auxiliary task given by the SHOT objective computed on clean data improves the results significantly ($> 4\%$). Further, we combine the aforementioned approach with the filtering approach of not updating the model on unreliable examples where we observe another performance boost of 1%. At last, we replace SHOT as an auxiliary task with our proposed distillation scheme in Section 2.2, while maintaining the SHOT objective on corrupted data. In this case, we observe another significant performance boost, corroborating the superiority of our proposed auxiliary task and the orthogonality of our components to the adaptation method.

### B.3.4 COMPONENTS OF DISTA

At last, we ablate the effect of each component of DISTA on the performance gain. Note that DISTA is reduced to EATA if we remove the proposed auxiliary task. To that end, we report in Table 11 the error rate of EATA, and its enhanced version through our proposed auxiliary task. Fist, we analyze the effect of introducing our distillation scheme via Cross Entropy (CE) on clean data without filtering. We observe a 0.5% reduction in the average error rate, with the performance gain reaching 0.8% on the motion blur corruption. Further, we analyze combining the aforementioned approach with filtering unreliable samples (by employing $\lambda_s(x_s)$), observing another 0.4% performance boost. Finally, we include sample reweighing and increase the filtering margin $E_0$ to $0.5\log(1000)$ resulting in another boost in accuracy (reduction in error rate). We note that we set the best hyperparameters for EATA, as recommended by the authors, with $E_0 = 0.4\log(1000)$.

Table 11: **Ablating DISTA with Episodic Evaluation on ImageNet-C with ResNet-50.** We ablate each component of DISTA where (CE) represents the distillation via Cross Entropy, (Fil) represents the filtering, and DISTA is the an improved version with better hyperparameter (setting $E_0 = 0.5\log(1000)$. Note that each proposed component provides a consistent performance boost.

| | Noise | | | Blur | | | | Weather | | | | Digital | | | | |
|---|---|---|---|---|---|---|---|---|---|---|---|---|---|---|---|---|
| | Gauss | Shot | Impul | Defoc | Glass | Motion | Zoom | Snow | Frost | Fog | Bright | Contr | Elastic | Pixel | Jpeg | Avg. |
| EATA | 64.0 | 62.1 | 62.5 | 66.9 | 66.9 | 52.5 | 47.4 | 48.2 | 54.2 | 40.2 | 32.2 | 54.6 | 42.2 | 39.2 | 44.7 | 51.9 |
| + CE | 63.2 | 61.2 | 61.6 | 66.3 | 66.3 | 51.7 | 46.9 | 47.9 | 53.9 | 39.7 | 31.9 | 54.3 | 41.9 | 39.1 | 44.4 | 51.4 |
| + Fil. | 62.9 | 60.7 | 61.4 | 65.8 | 65.9 | 51.2 | 46.5 | 47.6 | 53.7 | 39.3 | 31.7 | 54.3 | 41.6 | 38.5 | 44.1 | 51.0 |
| DISTA | **62.2** | **59.9** | **60.6** | **65.3** | **65.3** | **50.4** | **46.2** | **46.6** | **53.1** | **38.7** | **31.7** | **53.2** | **40.8** | **38.1** | **43.5** | **50.4** |

Table 12: **Effect of the Size of $\mathcal{D}_s$.** We report the error rate of DISTA under episodic evaluation on ImageNet-C when $\mathcal{D}_s$ is a sub-sampled set of the validation set of ImageNet. We observe that DISTA is robust under varying the size of $\mathcal{D}_s$. 'Ratio' represents the sub-sampled coefficient (*i.e.* ratio of 0.25 means that DISTA only leverages 25% of the validation set as $\mathcal{D}_s$).

| Ratio (%) | Noise | | | Blur | | | | Weather | | | | Digital | | | | Avg. |
|---|---|---|---|---|---|---|---|---|---|---|---|---|---|---|---|---|
| | Gauss | Shot | Impul | Defoc | Glass | Motion | Zoom | Snow | Frost | Fog | Bright | Contr | Elastic | Pixel | Jpeg | |
| EATA(0.0%) | 64.0 | 62.1 | 62.5 | 66.9 | 66.9 | 52.5 | 47.4 | 48.2 | 54.2 | 40.2 | 32.2 | 54.6 | 42.2 | 39.2 | 44.7 | 51.9 |
| DISTA(1.0%) | 63.1 | 61.1 | 61.1 | 66.7 | 65.8 | 50.9 | 46.7 | 47.3 | 53.7 | 39.1 | 31.9 | 54.1 | 41.5 | 38.6 | 44.1 | 51.1 |
| DISTA(2.5%) | 62.6 | 60.8 | 60.9 | 65.7 | 65.8 | 50.9 | 46.6 | 47.2 | 53.4 | 39.1 | 31.7 | 54.0 | 41.5 | 38.7 | 43.8 | 50.8 |
| DISTA(5.0%) | 62.4 | 60.4 | 60.9 | 65.5 | 66.0 | 50.5 | 46.3 | 46.9 | 53.2 | 38.9 | 31.8 | 53.6 | 41.0 | 38.3 | 43.8 | 50.6 |
| DISTA(7.5%) | 62.6 | 60.3 | 60.8 | 65.4 | 65.3 | 50.4 | 46.4 | 46.8 | 53.3 | 38.9 | 31.7 | 53.8 | 41.2 | 38.2 | 43.7 | 50.6 |
| DISTA(10%) | 62.4 | 60.3 | 60.2 | 65.5 | 65.5 | 50.6 | 46.3 | 46.7 | 53.1 | 38.8 | 31.7 | 53.5 | 41.1 | 38.2 | 43.8 | 50.5 |
| DISTA(25%) | 62.2 | 60.4 | 60.6 | 65.8 | 65.5 | 50.5 | 46.3 | 46.7 | 53.1 | 38.6 | 31.7 | 53.3 | 40.9 | 38.2 | 43.6 | 50.5 |
| DISTA(50%) | 62.3 | 60.4 | 60.4 | 65.1 | 65.7 | 50.6 | 46.2 | 46.7 | 53.3 | 38.7 | 31.7 | 53.2 | 40.9 | 38.3 | 43.4 | 50.5 |
| DISTA(75%) | 62.3 | 59.9 | 60.5 | 64.8 | 65.2 | 50.4 | 46.0 | 46.8 | 53.1 | 38.7 | 31.7 | 53.7 | 40.9 | 38.1 | 43.5 | 50.4 |
| DISTA(100%) | 62.2 | 59.9 | 60.6 | 65.3 | 65.3 | 50.4 | 46.2 | 46.6 | 53.1 | 38.7 | 31.7 | 53.2 | 40.8 | 38.1 | 43.5 | 50.4 |

## B.4 ABLATING THE SIZE OF $\mathcal{D}_s$

We complement our results with an ablation study on the effect of the size of source dataset $\mathcal{D}_s$ on the performance of DISTA. To that end, let $\mathcal{D}_s$ be a random subset of the validation set (unlabeled images). We conduct episodic evaluation on ImageNet-C using ResNet-50 dataset for this ablation and report the results in Table 12, where we observe DISTA is robust against variations in the size of $\mathcal{D}_s$. In particular, we observe that even with 10% of the validation set (*i.e.* storing 5000 unlabeled images), DISTA improves over EATA by 1.4% on average across all corruptions. Furthermore, with only 1% of the validation dataset (500 unlabeled images), DISTA still improves on EATA by 1% on shot and impulse noise.

## B.5 LIMITATIONS OF DISTA

In our experiments, we showed how DISTA is effective in multiple evaluation protocols, two datasets, and four different architectures. We note here that the performance improvement of DISTA comes at the cost of a memory burden (storing data samples from $\mathcal{D}_s$). However, our experiments in Table 12 shows that even with storing a very small set of unlabeled examples, DISTA is still effective in improving the performance.

## B.6 LEVERAGING LABELED SOURCE DATA

At last, we study a variation of DISTA for when labeled data from the source distribution is available. In this setting, one could replace the distillation loss in Equation equation 4 with a supervised loss function. To that end, we analyze one variant of DISTA where we replace the distillation loss with cross entropy loss between the prediction of $f_{\theta_t}$ and the ground-truth labels. The modified objective function can be expressed as:

$$\min_\theta \mathbb{E}_{x_t \sim \mathcal{S}} \lambda_t(x_t) E\left(f_\theta(x_t)\right) \ + \ \mathbb{E}_{(x_s, y_s) \sim \mathcal{D}_s} \lambda_s(x_s) CE\left(f_\theta(x_s), y_s\right)$$

We experiment with this labeled variant of DISTA and report the results on ImageNet-C in Table 13 under episodic evaluation using ResNet-50 architecture. We observe that leveraging hard (ground-truth) labels does not improve the result over our unsupervised distillation loss. Nevertheless, this supervised variant enhances the performance over the previous state-of-the-art method, EATA.

Table 13: **Episodic Evaluation on ImageNet-C Benchmark.** We compare the performance of EATA, DISTA, and leveraging labeled data for DISTA instead of the distillation task. We replace the distillation task with a cross entropy loss between the predictions and the ground-truth labels. We observe that our unsupervised distillation scheme outperforms both EATA and leveraging labeled data. Nevertheless, DISTA+Labeled still outperforms EATA by 0.8% on average.

| | Noise | | | Blur | | | | Weather | | | | Digital | | | | Avg. |
|---|---|---|---|---|---|---|---|---|---|---|---|---|---|---|---|---|
| | Gauss | Shot | Impul | Defoc | Glass | Motion | Zoom | Snow | Frost | Fog | Bright | Contr | Elastic | Pixel | Jpeg | |
| EATA | 64.0 | 62.1 | 62.5 | 66.9 | 66.9 | 52.5 | 47.4 | 48.2 | 54.2 | 40.2 | 32.2 | 54.6 | 42.2 | 39.2 | 44.7 | 51.9 |
| DISTA + Labels | 62.7 | 60.9 | 60.9 | 66.0 | 66.1 | 50.7 | 46.9 | 47.4 | 53.6 | 39.2 | 31.9 | 54.9 | 41.5 | 38.6 | 44.2 | 51.0 |
| DISTA | 62.2 | 59.9 | 60.6 | 65.3 | 65.3 | 50.4 | 46.2 | 46.6 | 53.1 | 38.7 | 31.7 | 53.2 | 40.8 | 38.1 | 43.5 | 50.4 |

