# OpenReview forum: "Test Time Adaptation with Auxiliary Tasks"
_ICLR.cc/2024/Conference — Submitted to ICLR 2024_

### Official Review · Reviewer_A87N · 2023-10-29

**Soundness:** 2 fair
**Presentation:** 3 good
**Contribution:** 2 fair
**Rating:** 5
**Confidence:** 3

**Summary:**

Post rebuttal: Some of the original concerns remain. The new experiments on labeled vs unlabelled source data further show that the narrative of the paper needs further work.

-------------------------------------------
The paper proposes a method for test-time adaptation that deals with limited data. This is done by adding a distillation loss on the source trained and frozen model on source data to the traditional entropy penalty on the target data. With experiments on ImageNet-C and ImageNet-3DCC, the paper shows that the method outperforms some of the recent techniques on TTA on online, continual and federated evaluations. The paper also proposes an analysis tool that is called _lookahead analysis_ which compares the entropy of predictions on a batch before and after doing the adaptation with the source distillation loss.

**Strengths:**

The proposed method is a simple extension to the existing methods, and is intuitive to understand. The experiments are sufficient to show the performance benefits. The work is placed well among existing works.

**Weaknesses:**

The method proposed in the paper is interesting, but the overall paper read like ad-hoc arguments stitched together. For example, the arguments for availability of _unlabled_ source data at inference, as opposed to storing some training data in a replay buffer is very unconvincing. It is also very unclear if the two problems that the paper set out to solve are tackled, as there doesn't seem to be any quantification of either one. The justification for federated evaluation are not convincing. Additionally, the lookahead analysis tool only measures the correlation between entropy of aux task and test data, but doesn't inform us of the test accuracy. While prior works have shows the correlation between test data entropy and accuracy, this cannot be taken for granted, as the paper doesn't show when the proposed method _doesn't_ work i.e., when is having source distillation not useful or harmless.

**Questions:**

In addition to the weaknesses mentioned, I have the following questions about the experimentation:
* The ablation only shows the batch size more than 8. How does the method behave when the batch size is 1. MEMO shows this is possible with special handling of batchnorm params.
* What are the effects of the source data batch size, and size of the unlabeled source data stored on performance?
* What are the trainable parameters? Is it only normalization or the whole network? What are the effects?
* What is the importance of the $E_0$ parameter on the data filtration process? While this is  proposed in prior works, an study of this in the current work could be illuminating.

If the authors can provide convincing arguments, I would be happy to raise my scores.

---

> ### Author Response · Authors · 2023-11-16
> **Official Comment by Authors**
>
> We thank the reviewer for their constructive comments. Next, we address the questions and weaknesses raised by the reviewer.
>
> **Regarding the use of unlabeled data instead of storing labeled examples in a replay buffer**: We thank the reviewer for this remark. While we agree that in some applications, storing labeled training examples is feasible, in other scenarios (such as CLIP [A]), labeled training data can be a private property that cannot be accessed. This was the main rationale behind using unlabeled data for our DISTA. Nevertheless, and as per the reviewers request, we conducted experiments on DISTA with replacing the distillation loss with a cross entropy loss with the ground truth labels. The overall objective function for DISTA-Labeled is:
>
> $\min_{\theta}  \underset{x_t\sim \mathcal S}{\mathbb E} \lambda_t(x_t) E\left(f_\theta(x_t)\right)  +   \underset{(x_s, y_s) \sim  \mathcal D_s}{\mathbb E} \lambda_s(x_s) CE\left(f_\theta(x_s),y_s\right)$
>
> Our results on ImageNet-C benchmark are in the table below where we report error rates.
>
> |            	| Gauss | Shot  | Impul | Defoc | Glass | Motion| Zoom  | Snow  | Frost | Fog   | Bright| Contr | Elastic | Pixel | Jpeg  | Avg.  |
> |----------------|-------|-------|-------|-------|-------|-------|-------|-------|-------|-------|-------|-------|---------|-------|-------|---------|
> | EATA       	| 64.0  | 62.1  | 62.5  | 66.9  | 66.9  | 52.5  | 47.4  | 48.2  | 54.2  | 40.2  | 32.2  | 54.6  | 42.2	| 39.2  | 44.7  | 51.9	|
> | DISTA + Labels | 62.7  | 60.9  | 60.9  | 66.0  | 66.1  | 50.7  | 46.9  | 47.4  | 53.6  | 39.2  | 31.9  | 54.9  | 41.5	| 38.6  | 44.2  | 51.0	|
> | DISTA      	| 62.2  | 59.9  | 60.6  | 65.3  | 65.3  | 50.4  | 46.2  | 46.6  | 53.1  | 38.7  | 31.7  | 53.2  | 40.8	| 38.1  | 43.5  | 50.4	|
>
> We observe that incorporating labeled data in DISTA instead of our distillation loss does not improve its results over using an unsupervised objective (while still providing state-of-the-art results compared to other baselines in our paper). We included this experiment in Appendix B.5 and referenced it in the main paper.
>
> **Regarding the justification for the federated evaluation**: We thank the reviewer for this remark. We see three main federated adaptation scenarios: (1) All clients experience the same distribution shift (e.g. fog). This setup would amount to a straight-forward application of federated learning methods to TTA, and thus is not the most interesting. (2) Different clients observe completely different distribution shifts, which is the most challenging setting; (3) Different clients experience different domain shifts that share some similarities or, as in our experiments, come from the same category (“setup” in our paper). We are tackling this latter scenario as a first step towards the most general case in (2). Besides being a relevant stepping stone towards more challenging federated settings, scenario (3) is also interesting in and of itself and allowed us to demonstrate that federated averaging still accelerates adaptation even when clients experience different domain shifts.
>
> **Regarding the lookahead analysis**: We appreciate the reviewer’s comment on the lookahead analysis. While indeed, one should quantify the lookahead based on performance measures  (e.g. accuracy), the labels at test time are not provided by the stream. We emphasize that this is an inherent challenge in the test-time-adaptation problem where the model is required to predict and leverage *unlabeled* data for the adaptation strategy. Hence, and following the TTA setup, we extended the lookahead analysis to this unsupervised setup by defining it in terms of entropy as a proxy metric, that is known to be highly correlated to test accuracy.
> We believe that providing the lookahead in terms of accuracy might make the analysis less practical for the TTA setup where labels are never available.
>
> **Regarding experiments with batch size smaller than 8**: We thank the reviewer for this comment. We note that we included experiments under batch size 1 in Table 9 in the appendix, where we compare DISTA against the state-of-the-art SAR under that setup. DISTA provided significant performance gains (around 5% reduction in error rate).
>
> **Regarding the effect of source data batch size**: We thank the reviewer for this comment. Figure 2 (a) in the main paper shows exactly this effect varying the source data batch size (curve in orange color). We note that DISTA is robust to such variations as we see that, for example, even with 50% less source data (50% smaller batch size or 50% less frequent updates on source data), DISTA achieves an average error rate on ImageNet-C that is 1.4% percentage lower than that of EATA.

---

> > ### Comment · Reviewer_A87N · 2023-11-22
> > **Reply to your rebuttal**
> >
> > Thank you for the rebuttal and the additional experiments.
> > I have a few more questions: In the Dista + labels results, you show that using CE over labels is worse than using the entropy. Why is that? Also, why can the entropy over source data still improve, assuming the source model has been trained _enough_?
> > Because of this, the lookahead analysis can be questioned that minimizing the source entropy is useful because the source training isn't enough (under a rather vague definition of the enough).
> >
> > In Part 2: Using just 1% of data, DISTA does well. Why do you think that is? And then, the benefits taper very quickly. Can you comment on this?

---

> ### Author Response · Authors · 2023-11-16
> **Official Comment by Authors (Part 2)**
>
> **Regarding the effect of the size of unlabeled source data stored on performance**: We thank the reviewer for suggesting this experimental ablation. Based on the reviewer’s suggestion, we conducted experiments where we reduced the size of source data used for DISTA. The Table below shows the results on episodic evaluation on ImageNet-C where DISTA leverages different fractions of the validation set of ImageNet sampled uniformly at random (without labels).
>
> | Method (Fraction)       	| Gauss 	| Shot  	| Impul 	| Defoc	| Glass	| Motion   | Zoom   | Snow   | Frost | Fog	| Bright   | Contr  | Elastic | Pixel  | Jpeg   | Avg.   |
> |-----------------------------|:---------:|:---------:|:---------:|:--------:|:--------:|:--------:|:------:|:------:|:-----:|:------:|:--------:|:------:|:-------:|:------:|:------:|:--------:|
> | EATA(0.0\%)             	| 64.0  	| 62.1  	| 62.5  	| 66.9 	| 66.9 	| 52.5 	| 47.4   | 48.2   | 54.2  | 40.2   | 32.2 	| 54.6   | 42.2	| 39.2   | 44.7   | 51.9 	|
> | DISTA(1.0\%)            	| 63.1  	| 61.1  	| 61.1  	| 66.7 	| 65.8 	| 50.9 	| 46.7   | 47.3   | 53.7  | 39.1   | 31.9 	| 54.1   | 41.5	| 38.6   | 44.1   | 51.1 	|
> | DISTA(2.5\%)            	| 62.6  	| 60.8  	| 60.9  	| 65.7 	| 65.8 	| 50.9 	| 46.6   | 47.2   | 53.4  | 39.1   | 31.7 	| 54.0   | 41.5	| 38.7   | 43.8   | 50.8 	|
> | DISTA(5.0\%)            	| 62.4  	| 60.4  	| 60.9  	| 65.5 	| 66.0 	| 50.5 	| 46.3   | 46.9   | 53.2  | 38.9   | 31.8 	| 53.6   | 41.0	| 38.3   | 43.8   | 50.6 	|
> | DISTA(7.5\%)            	| 62.6  	| 60.3  	| 60.8  	| 65.4 	| 65.3 	| 50.4 	| 46.4   | 46.8   | 53.3  | 38.9   | 31.7 	| 53.8   | 41.2	| 38.2   | 43.7   | 50.6 	|
> | DISTA(10\%)             	| 62.4  	| 60.3  	| 60.2  	| 65.5 	| 65.5 	| 50.6 	| 46.3   | 46.7   | 53.1  | 38.8   | 31.7 	| 53.5   | 41.1	| 38.2   | 43.8   | 50.5 	|
> | DISTA(25\%)             	| 62.2  	| 60.4  	| 60.6  	| 65.8 	| 65.5 	| 50.5 	| 46.3   | 46.7   | 53.1  | 38.6   | 31.7 	| 53.3   | 40.9	| 38.2   | 43.6   | 50.5 	|
> | DISTA(50\%)             	| 62.3  	| 60.4  	| 60.4  	| 65.1 	| 65.7 	| 50.6 	| 46.2   | 46.7   | 53.3  | 38.7   | 31.7 	| 53.2   | 40.9	| 38.3   | 43.4   | 50.5 	|
> | DISTA(75\%)             	| 62.3  	| 59.9  	| 60.5  	| 64.8 	| 65.2 	| 50.4 	| 46.0   | 46.8   | 53.1  | 38.7   | 31.7 	| 53.7   | 40.9	| 38.1   | 43.5   | 50.4 	|
> | DISTA(100\%)            	| 62.2  	| 59.9  	| 60.6  	| 65.3 	| 65.3 	| 50.4 	| 46.2   | 46.6   | 53.1  | 38.7   | 31.7 	| 53.2   | 40.8	| 38.1   | 43.5   | 50.4 	|
>
> We observe that DISTA is very robots against the size of $\mathcal D_s$. In particular, we find that even with 10\% of the validation set (\emph{i.e.} storing 5000 unlabeled images), DISTA improves over EATA by 1.4\% on average across all corruptions. Furthermore, with even (1\%) of source data, DISTA improves over EATA by 1\% on shot and impulse noise. This table along with its discussion is now at Appendix B.4.
>
> **Regarding the trainable parameters**: We thank the reviewer for this comment. As mentioned in appendix B.1, DISTA only updates the learnable parameters of normalization layers. This choice is common practice in the TTA literature [A, B, C]. We agree this is an important detail and decided to move this experimental detail to the main paper for better clarity.
>
> **Regarding the importance of $E_0$**: We appreciate the reviewer’s comment. E_0 plays a vital role in filtering unreliable samples (samples with high entropy). We conducted  experiments with two values of E_0 (0.4*log(1000), 0.5*log(1000)) and report the results in the table below.
>
> | E0   | Gauss     	| Shot      	| Impul     	| Defoc     	| Glass     	| Motion    	| Zoom      	| Snow      	| Frost     	| Fog       	| Bright    	| Contr     	| Elastic   	| Pixel     	| Jpeg      	| Avg.      	|
> |----------|---------------|---------------|---------------|---------------|---------------|---------------|---------------|---------------|---------------|---------------|---------------|---------------|---------------|---------------|---------------|---------------|
> | 0.4 log(1000)   | 62.9      	| 60.7      	| 61.4      	| 65.8      	| 65.9      	| 51.2      	| 46.5      	| 47.6      	| 53.7      	| 39.3      	| 31.7      	| 54.3      	| 41.6      	| 38.5      	| 44.1      	| 51.0      	|
> | 0.5 log(1000)	| 62.2      	| 59.9      	| 60.6      	| 65.3      	| 65.3      	| 50.4      	| 46.2      	| 46.6      	| 53.1      	| 38.7      	| 31.7      	| 53.2      	| 40.8      	| 38.1      	| 43.5	| 50.4
>
>
> We observe that varying E_0 results in performance variation due to filtering more/less samples from the update step. This result is in Table 11 in the Appendix.
>
>
> We hope that our answers addressed the reviewers weaknesses and questions. We are happy to engage in further discussion to clarify any confusion or misunderstanding.

---

> ### Author Response · Authors · 2023-11-22
> **Reply to Post-Rebuttal Feedback**
>
> We thank the reviewer for the additional feedback.
>
> **Regarding the performance of DISTA+Labels**: We note first, that DISTA + Labels is better than solely conducting entropy minimization (EATA). That is, including the additional supervised auxiliary task of leveraging source data is beneficial to the adaptation task (reduced error rate from 51.9 to 51.0 on average on ImageNet-C). Thus, we suspect that the supervised loss will have a positive lookahead. Due to the very narrow time limit until the end of the discussion period, we are unable to provide this lookahead analysis in our response. We commit to include the lookahead plots for the DISTA+labels baseline in the final version of our work.
>
> Note that, However, DISTA+Labels performs worse than DISTA. We believe that the supervised loss that includes labels adds an additional complexity to the adaptation due to the imperfect performance of the trained models. That is, if a pretrained model is having a perfect performance with perfect calibration (i.e. predicting all samples correctly, and the correct class is assigned a probability of 1), then DISTA+Labels is reduced to DISTA.
> We hypothesize that the benefit of using source data in DISTA is that it regularizes the test-time adapted model to remain close to the original one in functional space.
> This probably eases the TTA optimization problem, since in the absence of labels for the test data, TTA methods  have to resort to noisy proxies to the underlying classification objective, which might lead the model away from the original classifier and into poor local minima. In that light, it is reasonable to expect that the output of the original model output makes for a better regularization than the labels themselves.
>
> **Regarding the improvement that entropy minimization on source data can provide (Equation 2)**: We hypothesize that during the adaptation process, the adapted model overfits to the presented distribution from the stream while increasing its uncertainty on the clean data. However, including a simple entropy minimization on source data might mitigate partially this effect by providing a more stable adaptation process. We note that we confirmed the usefulness of this approach in Table 6 in the main paper where Aux-Tent improved the error rate over Tent by more than 1% on Gaussian, Shot, and Impulse noises, and by 0.6% on average on ImageNet-C confirming the earlier lookahead analysis in Figure 1 (a).
>
> **Regarding the effectiveness of using small portions of source data as $\mathcal D_s$**: Coming back to the idea of regularization mentioned above, the intuition behind the usage of source data is not necessarily to maintain the best performance possible on the original dataset but to prevent the test-time-adapted model to diverge too much from the original classifier. Arguably, the latter is a much simpler task, and it seems that 1% of the data (500 images in this case) already includes enough variability to regularize the model effectively. Adding more source data improves such a regularization, but the regularization alone cannot drive the model’s performance, which explains why we quickly reach a regime of diminishing returns in terms of the availability of source data.
>
> We hope that our answers addressed the reviewers questions. We are happy to engage in further discussion to clarify any confusion or misunderstanding.

---

### Official Review · Reviewer_XCKF · 2023-11-01

**Soundness:** 2 fair
**Presentation:** 3 good
**Contribution:** 2 fair
**Rating:** 6
**Confidence:** 4

**Summary:**

The paper focuses on improving test time adaptation of ImageNet pretrained models on distribution shifts of ImageNet-C and ImageNet-3DCC. In addition to the test time entropy minimization objective, the paper propose to distill the predictions of unlabeled source samples from the original pretrained model during adaptation. Authors show that this additional auxiliary task improves model adaptation on target distribution data in both episodic and continual test time adaptation settings.

**Strengths:**

-	Proposed a novel auxiliary task for test-time adaptation problem setting.

-	Proposed method is simple and shown to be effective.

-	Method explanation is easy to understand and follow.

-	Results show good improvement over prior work on both ImageNet-C and ImageNet-3DCC.

**Weaknesses:**

-	Intuition behind eq, (3) is not straightforward. In particular, $\theta_{t+1}$ is obtained by updating ${\theta_t}^c$ using the gradients w.r.t  $\theta$. Discussing the intuition behind it would be helpful.

-	It is mentioned in the 2nd line of Page 4, $\mathcal{D}_s$ is sampled from the training dataset. The pretrained network has seen the $\mathcal{D}_s$ samples during its pretraining and potentially have lower entropy on those samples. Following the motivation of this work that targets to use unlabeled data, it is important to use unseen unlabeled data from source distribution for the auxiliary tasks, not the samples from the seen training set.

-	No ablation study on the size of unlabeled source dataset $\mathcal{D}_s$ is provided. This ablation study is important as this dataset shown to guide the adaptation process.

-	category-wise federated TTA, where all clients are assumed to know beforehand that adaptation carries out on similar category of domain shifts. It is a controlled setting, and not a realistic one. Federated TTA with diverse distribution shifts across clients would be an interesting scenario.

**Questions:**

-	Consider sampling $\mathcal{D}_s$ from unseen unlabeled data from source distribution.
-	What should be the size of $\mathcal{D}_s$? How the results vary with the $\mathcal{D}_s$ size? Do the samples in $\mathcal{D}_s$ need to be class-balanced?
-	Is $f_\theta$ in eq. (3) defaults to $f_{\theta}$ at time t?
-	What is the intuition behind updating ${\theta_t}^c$ using the gradients w.r.t  $\theta$ in eq. (3)?
-	Since $\mathcal{D}_s$ are sampled from training set, how the results would change if ground-truth labels are used in eq. (5)?
-	I understood the lookahead concept. However, what makes entropy minimization on clean data accelerate adaptation. Data distribution between clean data and targeted adapted data is different, so no additional distribution knowledge is provided to the network.
-	Please mention the network architecture used in respective figure and table captions.

I will reconsider my rating based on the authors feedback on my concerns.

---

> ### Author Response · Authors · 2023-11-16
> **Official Comment by Authors**
>
> We thank the reviewer for their constructive comments. We are glad that the reviewer recognizes the novelty of our auxiliary task for test-time adaptation and the effectiveness of our method. Next, we address the questions and weaknesses raised by the reviewer.
>
> **Regarding the intuition behind Equation (3)**: The main intuition behind Equation (3) is to solve the optimization problem in Equation (2) in an alternating optimization manner. This is because we want to analyze the impact of the auxiliary task (second term in Equation 2), and conducting this alternating optimization approach grant us an access to the intermediate model parameters $\theta_t^c$ and allows us to compare the performances of $\theta_{t+1}$ and $\theta_t^c$. We edited the paragraph before equation (3) to clarify that. Note that the expectations in equation (3) are taken over different data distributions (resp. corrupted and clean data distributions), and while the gradients w.r.t. $\theta$ are used to update $\theta^c_t$ to get $\theta_{t+1}$, the gradient is evaluated at $\theta_t^c$. That is,
>
> $\theta_t^c = \theta_t - \gamma \nabla_\theta [E(f_\theta(x_t)]|_{\theta = \theta_t}$
>
> $\theta_{t+1} = \theta_t^c - \gamma \nabla_\theta [E(f_\theta(x_t)]|_{\theta = \theta_t^c}$
>
> We hope that this has clarified the intuition behind Equation (3).
>
> **Regarding the effect of the size of unlabeled source data stored on performance**: We thank the reviewer for suggesting this experimental ablation. Based on the reviewer’s suggestion, we conducted experiments where we reduced the size of source data used for DISTA. The Table below shows the results on episodic evaluation on ImageNet-C where DISTA leverages different fractions of the validation set of ImageNet sampled uniformly at random (without labels).
>
> | Method (Fraction)       	| Gauss 	| Shot  	| Impul 	| Defoc	| Glass	| Motion   | Zoom   | Snow   | Frost | Fog	| Bright   | Contr  | Elastic | Pixel  | Jpeg   | Avg.   |
> |-----------------------------|:---------:|:---------:|:---------:|:--------:|:--------:|:--------:|:------:|:------:|:-----:|:------:|:--------:|:------:|:-------:|:------:|:------:|:--------:|
> | EATA(0.0\%)             	| 64.0  	| 62.1  	| 62.5  	| 66.9 	| 66.9 	| 52.5 	| 47.4   | 48.2   | 54.2  | 40.2   | 32.2 	| 54.6   | 42.2	| 39.2   | 44.7   | 51.9 	|
> | DISTA(1.0\%)            	| 63.1  	| 61.1  	| 61.1  	| 66.7 	| 65.8 	| 50.9 	| 46.7   | 47.3   | 53.7  | 39.1   | 31.9 	| 54.1   | 41.5	| 38.6   | 44.1   | 51.1 	|
> | DISTA(2.5\%)            	| 62.6  	| 60.8  	| 60.9  	| 65.7 	| 65.8 	| 50.9 	| 46.6   | 47.2   | 53.4  | 39.1   | 31.7 	| 54.0   | 41.5	| 38.7   | 43.8   | 50.8 	|
> | DISTA(5.0\%)            	| 62.4  	| 60.4  	| 60.9  	| 65.5 	| 66.0 	| 50.5 	| 46.3   | 46.9   | 53.2  | 38.9   | 31.8 	| 53.6   | 41.0	| 38.3   | 43.8   | 50.6 	|
> | DISTA(7.5\%)            	| 62.6  	| 60.3  	| 60.8  	| 65.4 	| 65.3 	| 50.4 	| 46.4   | 46.8   | 53.3  | 38.9   | 31.7 	| 53.8   | 41.2	| 38.2   | 43.7   | 50.6 	|
> | DISTA(10\%)             	| 62.4  	| 60.3  	| 60.2  	| 65.5 	| 65.5 	| 50.6 	| 46.3   | 46.7   | 53.1  | 38.8   | 31.7 	| 53.5   | 41.1	| 38.2   | 43.8   | 50.5 	|
> | DISTA(25\%)             	| 62.2  	| 60.4  	| 60.6  	| 65.8 	| 65.5 	| 50.5 	| 46.3   | 46.7   | 53.1  | 38.6   | 31.7 	| 53.3   | 40.9	| 38.2   | 43.6   | 50.5 	|
> | DISTA(50\%)             	| 62.3  	| 60.4  	| 60.4  	| 65.1 	| 65.7 	| 50.6 	| 46.2   | 46.7   | 53.3  | 38.7   | 31.7 	| 53.2   | 40.9	| 38.3   | 43.4   | 50.5 	|
> | DISTA(75\%)             	| 62.3  	| 59.9  	| 60.5  	| 64.8 	| 65.2 	| 50.4 	| 46.0   | 46.8   | 53.1  | 38.7   | 31.7 	| 53.7   | 40.9	| 38.1   | 43.5   | 50.4 	|
> | DISTA(100\%)            	| 62.2  	| 59.9  	| 60.6  	| 65.3 	| 65.3 	| 50.4 	| 46.2   | 46.6   | 53.1  | 38.7   | 31.7 	| 53.2   | 40.8	| 38.1   | 43.5   | 50.4 	|
>
> We observe that DISTA is very robots against the size of $\mathcal D_s$. In particular, we find that even with 10\% of the validation set (i.e. storing 5000 unlabeled images), DISTA improves over EATA by 1.4\% on average across all corruptions. Furthermore, with even (1\%) of source data, DISTA improves over EATA by 1\% on shot and impulse noise. This table along with its discussion is now at Appendix B.4.

---

> ### Author Response · Authors · 2023-11-16
> **Official Comment by Authors (Part 2)**
>
> **Regarding the use of unseen unlabeled data from the source distribution**: We thank the reviewer for this great comment. Note that as mentioned in the footnote in page 6 that we also experimented with $\mathcal D_s$ being the validation set of ImageNet and did not notice any changes in our results. For completeness, we are attaching the results below. We report the results on ImageNet-C under episodic evaluation.
> |   Split    | Gauss | Shot | Impul | Defoc | Glass | Motion | Zoom | Snow | Frost | Fog | Bright | Contr | Elastic | Pixel | Jpeg | Avg.  |
> |-------|-------|------|-------|-------|-------|--------|------|------|-------|-----|--------|-------|---------|-------|------|-------|
> | Tain  | 62.3  | 60.1 | 60.8  | 65.3  | 65.4  | 50.3   | 46.1 | 46.8 | 53.1  | 38.7| 31.7   | 53.5  | 40.7    | 38.2  | 43.5 | 50.4  |
> | Val   | 62.2  | 59.9 | 60.6  | 65.3  | 65.3  | 50.4   | 46.2 | 46.6 | 53.1  | 38.7| 31.7   | 53.2  | 40.8    | 38.1  | 43.5 | 50.4  |
>
>
> **Regarding the federated evaluation**: We thank the reviewer for this remark. We see three main federated adaptation scenarios: (1) All clients experience the same distribution shift (e.g. fog). This setup would amount to a straight-forward application of federated learning methods to TTA, and thus is not the most interesting. (2) Different clients observe completely different distribution shifts, which is the most challenging setting; (3) Different clients experience different domain shifts that share some similarities or, as in our experiments, come from the same category (“setup” in our paper). We are tackling this latter scenario as a first step towards the most general case in (2). Besides being a relevant stepping stone towards more challenging federated settings, scenario (3) is also interesting in and of itself and allowed us to demonstrate that federated averaging is still useful even when clients experience different domain shifts. Since in TTA performance is measured in an online manner, the information shared via federated updates can significantly speed up adaptation.
>
>
> **Is $f_theta$ in eq. (3) defaults to $f_theta$  at time $t$?** In the first part of Equation 3, f_{\theta_t} is used to estimate the entropy, while in the right hand side, $f_{\theta_t^c}$ is used. We added a clarification description about Equation 3 in page 3.
>
> **Regarding the use of labeled data instead of distillation loss**: as per the reviewers request, we conducted experiments on DISTA with replacing the distillation loss with a cross entropy loss with the ground truth labels. The overall objective function for DISTA-Labeled is:
>
> $
> \min_{\theta}  \underset{x_t\sim \mathcal S}{\mathbb E} \lambda_t(x_t) E\left(f_\theta(x_t)\right)  +   \underset{(x_s, y_s) \sim  \mathcal D_s}{\mathbb E} \lambda_s(x_s) CE\left(f_\theta(x_s),y_s\right)
> $
>
> Our results on ImageNet-C benchmark are in the table below.
>
> |            	| Gauss | Shot  | Impul | Defoc | Glass | Motion| Zoom  | Snow  | Frost | Fog   | Bright| Contr | Elastic | Pixel | Jpeg  | Avg.  |
> |----------------|-------|-------|-------|-------|-------|-------|-------|-------|-------|-------|-------|-------|---------|-------|-------|---------|
> | EATA       	| 64.0  | 62.1  | 62.5  | 66.9  | 66.9  | 52.5  | 47.4  | 48.2  | 54.2  | 40.2  | 32.2  | 54.6  | 42.2	| 39.2  | 44.7  | 51.9	|
> | DISTA + Labels | 62.7  | 60.9  | 60.9  | 66.0  | 66.1  | 50.7  | 46.9  | 47.4  | 53.6  | 39.2  | 31.9  | 54.9  | 41.5	| 38.6  | 44.2  | 51.0	|
> | DISTA      	| 62.2  | 59.9  | 60.6  | 65.3  | 65.3  | 50.4  | 46.2  | 46.6  | 53.1  | 38.7  | 31.7  | 53.2  | 40.8	| 38.1  | 43.5  | 50.4	|
>
> We observe that incorporating labeled data in DISTA instead of our distillation loss does not improve its results over using an unsupervised objective (while still providing state-of-the-art results compared to other baselines in our paper). We included this experiment in Appendix B.5 and referenced it in the main paper.
>
> **Regarding mentioning the network architecture in captions**: as per the reviewers request, we updated the captions of all tables in the main paper and appendix to mention the network architecture used.
>
> We hope that our answers addressed the reviewers weaknesses and questions. We are happy to engage in further discussion to clarify any confusion or misunderstanding.

---

> ### Author Response · Authors · 2023-11-22
> **Last Day of Discussion**
>
> Dear Reviewer
>
> As the discussion period is about to end (22nd of November), we ask the reviewer to take a look at our response and modifications to the paper to address their questions and weaknesses. We are happy to engage further in discussions to clarify any confusion or misunderstanding.
>
> Best,

---

### Official Review · Reviewer_JXKk · 2023-11-01

**Soundness:** 2 fair
**Presentation:** 1 poor
**Contribution:** 2 fair
**Rating:** 3
**Confidence:** 5

**Summary:**

The authors propose to leverage source data to improve the performance of EATA, a state of the art test-time adaptation algorithm. They present results on ImageNet-C, and -3DCC for small sized models (Resnet18, 50, ViT). The proposed method outperforms the state of the art, EATA on most of these tasks.

**Strengths:**

- Clarity: Execution of experiments and write-up of the paper story is clear.

- Quality: Results are well evaluated, yet limited in the choice of models and datasets.

- Significance: Paper shows that combination of EATA and UDA setup improves over TTA methods, see concerns below.

General comment: The motivation for the paper is clear, and results are interesting and thorough (but see the weaknesses). Limited analyis on hyperparameter effects was done, but some is included. While the objective function is a combination of UDA and EATA, the adapting weighting of the source and target losses, Eq. 4, sounds like a novel contribution.

**Weaknesses:**

The authors essentially re-invented domain adaptation, and try to re-sell this as a SOTA TTA approach.

In the classical domain adaptation setting, entropy minimization or pseudo labeling methods are combined with a cross-entropy on the source dataset, see e.g. [French et al. (2017)](https://arxiv.org/abs/1706.05208). This is a very well-studied method, and established to work well.

In this regard, I find the paper story a bit problematic: The authors position themselves in the space of test-time adaptation, which is exactly the setting in which source data is *not available*. The fact that the state-of-the-art TTA model, EATA*, when evaluated in an unsupervised domain adaptation setting instead of a TTA setting becomes better might be obvious.

That being said, a well-made evaluation on applying domain adaptation techniques to TTA might be interesting to increase cross-talk between the two fields/evaluation settings, but this is completely dismissed by how this paper is written. There is not even a single instance (besides a few titles in the bibliography) mentioning the word "domain adaptation", or better, "unsupervised domain adaptation", in the whole paper.

I am not sure about the best way to rectify this, and would be happy to engage in a discussion with the authors. I would be especially keen to know:

Additional main weaknesses:

- I find it confusing that at first, entropy minimization on the source data is introduced, and then switched to cross-entropy. As I noted above, this is a very standard and established thing in the literature, and I would re-write this part of the methods section with that in mind.
- "auxiliary task": I would change the naming here, again, and note that this is unsupervised domain adaptation. The paper title is also very misleading, as it suggests that multiple "auxiliary task*ss*" are considered when it is in fact only one.
- Given my comments about unsupervised DA vs. TTA, an extensive review on SOTA UDA methods is lacking, and fair experimental evaluation towards these SOTA methods. If the authors are interested in rectifying this issue, I would be happy to discuss a set of methods to benchmark against before running experiments.
- There is no discussion on the additional memory burden introduced by storing the source data in the paper. In Figure 2, EATA and DISTA are compared in terms of additional computation, but not in terms of additional memory. The memory requirement is quite drastic: [ResNet50](https://pytorch.org/vision/main/models/generated/torchvision.models.resnet50.html) weights are quoted at about 97.8 MB per model. Storing the source model is easy (as batch norm params are light), but storing the *source dataset*, which is required for this method, is of course very memory heavy. Can the authors comments if the results were conducted with using the full imagenet dataset, or subsampling it somehow? If the full imagenet dataset is used, this adds 160GB of additional storage. At this storage requirement, it would have been possible to run much bigger models, e.g. an EffNet-L2 ([Xie et al., 2020](https://arxiv.org/abs/1911.04252)) which cuts the reported error rates much more drasticially than DISTA (under higher compute requirements, of course). The memory requirement is a big limitation, and (in my opinion) one of the main reason why TTA is hard.
- Why not consider datasets for natural corruptions, like ImageNet-R, -D, ObjectNet, etc., on top of ImageNet-C?

Additional weaknesses and comments:

- "signficiantly" is used throughout the paper, and not a single error bar is provided. Please either compute error bars and proper stats, or replace this by a different term.
- There is a confusion of "improvement by X%" when "improvement by X percentage points"/"X% points" is meant. I think the confusion is used consistently, though, so might be fine.
- "large scale": I find it questionable if nowadays, ImageNet-C still constitutes a large scale dataset, but this is a minor point/comment. I would suggest to simply drop the term.
- Methodology: The start is loaded with (in my opinion) unnecessary notation for very simple concepts like how the classifier is setup. I would suggest to make this more crisp, as it does not really add much to the paper, the setup is very standard.
- The table headings should outlined which model is used for the results
- How stochastic is the method? Errorbars should be provided.
- The empirical results are somewhat limited in the breadth of explored methods. It would improve/broaden the scope if additional models, e.g. larger resnets etc. were used.


___
* note that also EATA slightly deviates from the pure TTA setup, as they leveraged clean source images for the computation of their regularizer, cf. the original paper for details.

**Questions:**

1. Why is there no discussion of domain adaptation, given that Eq. (4) is pretty much exactly the commonly used formulation of unsupervised domain adaptation?
2. If access to source data is allowed, then a wealth of non-TTA, unsupervised domain adaptation methods exist to perform the adaptation task. However, the paper only benchmarks against TTA approaches. Which
3. Did you re-ran the results of other methods and yours within a single codebase (i.e., did you re-eval prev methods), or did you copy numbers from papers?
4. How were $\epsilon$ and $E_0$, which influence the weighting of both losses, tuned? An analyis table for varying these two parameters on a hold-out set (or however they were validated) is missing and should be added to the next paper version.


**Additional questions:**

- Abstract, "key challenge in TTA: adapting on limited data": Can you give references that identify this as a key challenge? Quite to the contrary, when adapting a model at test-time, especially in continual settings, I would argue that a wealth of data exists.
- Abstract, "conducting several adaptation steps ... will lead to overfitting": I am not aware of work suffering from this. Can you give a reference? Is there an experiment in the paper where you specifically show that DISTA resolves this?
- Abstract, in many realistic scenarios, the stream revals insufficent data to fully adapt": What are examples of such settings, and where was this shown?

---

> ### Author Response · Authors · 2023-11-16
> **Official Comment by Authors**
>
> We thank the reviewer for their valuable comments. We are glad that the reviewer recognizes the clarity and significance of our results. Next, we address the questions and weaknesses raised by the reviewer.
>
> **Regarding the relation between our work and unsupervised domain adaptation**: We appreciate the reviewer’s comment and the interesting link to unsupervised domain adaptation, however, we disagree about casting our work as a reinvention of domain adaptation. We would like to clarify first the following:
> Differences between TTA and Unsupervised Domain Adaptation: The reviewer pointed out that the difference between test-time adaptatio and unsupervised domain adaptation is the access to source data during adaptation; specifically that in TTA source data is assumed to be unavailable. We respectfully disagree with that view, and in fact we are not the first to propose a TTA method that leverages source data. RMT [A], EATA [B] and ActMAD [C] were developed under the premise that source data is available and still are considered integral parts of the TTA literature. More recently, Kang et.al. [D] distilled source data for the sake of adaptation, as mentioned in our related work. All these examples demonstrate that the use of source data is accepted and not uncommon in the Test-Time Adaptation literature. While we do agree that there are similarities between TTA and UDA, we believe there are key differences in (1) the way the pretrained model is accessing the target distribution(s), and (2) the way an adaptation method is evaluated, as we elaborate below.
> (1) In Domain adaptation, the pretrained model can access the target domain *all at once* for adaptation. However, in Test-Time Adaptation, the pretrained model accesses the target domain as a stream of data. The model observes each revealed batch of data only *once* unlike domain adaptation where one could potentially revisit examples from the target domain multiple times.
> (2) The performance in domain adaptation is calculated *after* adaptation. That is, after a pretrained model adapts on a target domain, its performance on novel samples from the target domain is calculated. In Test-Time Adaptation, the performance of an adaptation method is calculated in an *online* manner where the model needs to provide the prediction of a given batch before receiving the next batch from the stream (please refer to the preliminary paragraph in our method section where we formalize the interaction between a TTA method and the stream of unlabeled data).
> Given these differences, we believe that our work, similar to [A, B, C, D], falls into the test time adaptation category where we followed the standard evaluation setups in the literature.
> While we do believe that cross-talk between the two fields is very interesting and important, we think that this deserves a work on its own and it falls outside the scope of our submission.Nonetheless, we are happy to include a discussion and comparison (if possible) on related unsupervised domain adaptation methods that the reviewer thinks are most related.
>
> [A] Robust Mean Teacher for Continual and Gradual Test-Time Adaptation, CVPR 2023
>
> [B] Efficient Test-Time Model Adaptation without Forgetting, ICML 2022
>
> [C] ActMAD: Activation Matching To Align Distributions for Test-Time-Training, CVPR2023
>
> [D] Leveraging proxy of training data for test-time adaptation, ICML 2023.
>
>
> **Regarding the confusion due to the use of “auxiliary tasks”**: Response: In our work, we analyzed the effectiveness of 3 different auxiliary tasks. Equation (2) shows a simple entropy minimization on source data as auxiliary task while Equation (4) shows our proposed distillation-based auxiliary task. In Table 6, we reported the performance of Aux-SHOT which employs an information maximization on source data as an auxiliary task. Moreover, auxiliary tasks have proven useful in numerous machine learning applications, and we believe formalizing our contributions as auxiliary tasks might facilitate and encourage the study of new auxiliary tasks in TTA

---

> > ### Author Response · Authors · 2023-11-16
> > **Official Comment by Authors (Part 2)**
> >
> > **Regarding the discussion on memory burden**: We would like to thank the reviewer for this comment. We would like to point out that DISTA uses the same amount of source data as EATA uses for calculating the Fisher matrix. In particular, DISTA leverages a subset of (50,000) unlabeled images which translates to around 4% of ImageNet training set in size. Nevertheless, we do agree that the performance improvement of DISTA comes at an additional storage cost, similar to [A, B, C, D]. We have added a limitation section in our appendix discussing the additional memory requirements of DISTA. Further, we conducted experiments where we use subsample the ImageNet validation set (unlabeled) and use it as  $\mathcal D_s$ to reduce the memory burden. Here are the results:
> > | Method (Fraction)       	| Gauss 	| Shot  	| Impul 	| Defoc	| Glass	| Motion   | Zoom   | Snow   | Frost | Fog	| Bright   | Contr  | Elastic | Pixel  | Jpeg   | Avg.   |
> > |-----------------------------|:---------:|:---------:|:---------:|:--------:|:--------:|:--------:|:------:|:------:|:-----:|:------:|:--------:|:------:|:-------:|:------:|:------:|:--------:|
> > | EATA(0.0\%)             	| 64.0  	| 62.1  	| 62.5  	| 66.9 	| 66.9 	| 52.5 	| 47.4   | 48.2   | 54.2  | 40.2   | 32.2 	| 54.6   | 42.2	| 39.2   | 44.7   | 51.9 	|
> > | DISTA(1.0\%)            	| 63.1  	| 61.1  	| 61.1  	| 66.7 	| 65.8 	| 50.9 	| 46.7   | 47.3   | 53.7  | 39.1   | 31.9 	| 54.1   | 41.5	| 38.6   | 44.1   | 51.1 	|
> > | DISTA(2.5\%)            	| 62.6  	| 60.8  	| 60.9  	| 65.7 	| 65.8 	| 50.9 	| 46.6   | 47.2   | 53.4  | 39.1   | 31.7 	| 54.0   | 41.5	| 38.7   | 43.8   | 50.8 	|
> > | DISTA(5.0\%)            	| 62.4  	| 60.4  	| 60.9  	| 65.5 	| 66.0 	| 50.5 	| 46.3   | 46.9   | 53.2  | 38.9   | 31.8 	| 53.6   | 41.0	| 38.3   | 43.8   | 50.6 	|
> > | DISTA(7.5\%)            	| 62.6  	| 60.3  	| 60.8  	| 65.4 	| 65.3 	| 50.4 	| 46.4   | 46.8   | 53.3  | 38.9   | 31.7 	| 53.8   | 41.2	| 38.2   | 43.7   | 50.6 	|
> > | DISTA(10\%)             	| 62.4  	| 60.3  	| 60.2  	| 65.5 	| 65.5 	| 50.6 	| 46.3   | 46.7   | 53.1  | 38.8   | 31.7 	| 53.5   | 41.1	| 38.2   | 43.8   | 50.5 	|
> > | DISTA(25\%)             	| 62.2  	| 60.4  	| 60.6  	| 65.8 	| 65.5 	| 50.5 	| 46.3   | 46.7   | 53.1  | 38.6   | 31.7 	| 53.3   | 40.9	| 38.2   | 43.6   | 50.5 	|
> > | DISTA(50\%)             	| 62.3  	| 60.4  	| 60.4  	| 65.1 	| 65.7 	| 50.6 	| 46.2   | 46.7   | 53.3  | 38.7   | 31.7 	| 53.2   | 40.9	| 38.3   | 43.4   | 50.5 	|
> > | DISTA(75\%)             	| 62.3  	| 59.9  	| 60.5  	| 64.8 	| 65.2 	| 50.4 	| 46.0   | 46.8   | 53.1  | 38.7   | 31.7 	| 53.7   | 40.9	| 38.1   | 43.5   | 50.4 	|
> > | DISTA(100\%)            	| 62.2  	| 59.9  	| 60.6  	| 65.3 	| 65.3 	| 50.4 	| 46.2   | 46.6   | 53.1  | 38.7   | 31.7 	| 53.2   | 40.8	| 38.1   | 43.5   | 50.4 	|
> >
> > We observe that  even with 10\% of the validation set (\emph{i.e.} storing 5000 unlabeled images), DISTA improves over EATA by 1.4\% on average across all corruptions. Furthermore, with even (1\%) of source data, DISTA improves over EATA by 1\% on shot and impulse noise. We added this experiment with its discussion in Table 12 (Appendix B.4).
> > Above all,, it is noteworthy that the cost of storage is generally much cheaper than the cost of computation. For instance, based on Table 3 in [E] (Appendix A), storing the entire ImageNet 1K dataset costs 66 cents which is much cheaper than training a single model on the same dataset (cost of 500 USD).  That is, storing more data is cheaper than leveraging larger models for inference.
> >
> >
> > [E] Computationally Budgeted Continual Learning: What Does Matter?, CVPR2023
> >
> > **Regarding conducting experiments on more datasets**: We thank the reviewer for this suggestion. In our paper, we followed the common practice in the TTA literature [A, B, F, G]  and  conducted experiments on the two largest  TTA benchmarks: ImageNet-C and ImageNet-3DCC
> >
> > [F] Tent: Fully Test-time Adaptation by Entropy Minimization, ICLR 2021
> >
> > [G] Towards Stable Test-Time Adaptation in Dynamic Wild World, ICLR 2023
> >
> >
> > **Regarding the use of “significantly”**: We thank the reviewer for this comment. We note here that ‘significantly’ here does not refer to statistical significance, but rather to a ‘notable’ performance gain. If this is still a source of confusion, we are happy to repeat each experiment multiple times and report the statistics.
> >
> > **Regarding mentioning the network architecture in captions**: We would like to thank the reviewer for this comment. We have modified the table captions to include the model used in the experiments.

---

> > ### Comment · Reviewer_JXKk · 2023-12-02
> > **Re: UDA vs. TTA**
> >
> > Thanks for the rebuttal. I considered your comments, but remain unconvinced, here is why:
> >
> > > We respectfully disagree with that view, and in fact we are not the first to propose a TTA method that leverages source data. RMT [A], EATA [B] and ActMAD [C] were developed under the premise that source data is available and still are considered integral parts of the TTA literature. More recently, Kang et.al. [D] distilled source data for the sake of adaptation, as mentioned in our related work.
> >
> > I think this is not correct, please refer to Table 1 in EATA for an overview. The proposed loss in this paper clearly falls into the category of "Unsupervised domain adaptation", or potentially also "Test time training" if you compare the form of training loss. The comparision to work that "distills" source information is correct, but here consider the difference in the memory footprint (full dataset in your case, vs. the cost of distillation in other cases).
> >
> > > (1) In Domain adaptation, the pretrained model can access the target domain all at once for adaptation. However, in Test-Time Adaptation, the pretrained model accesses the target domain as a stream of data. The model observes each revealed batch of data only once unlike domain adaptation where one could potentially revisit examples from the target domain multiple times
> >
> > I do not think that this argument would hinder you from benchmarking a state of the art UDA approach.
> >
> > The considered evaluation scheme in the paper also randomizes incoming data, so I do not see a conceptual difference in the evaluation scheme here. This argument would be more convincing for very slowly drifting distributions (e.g. class incremental learning, etc), but the evaluation scheme using ImageNet-C and other datasets in this study leverages randomized incoming data. So while interesting conceptually, the chosen evaluation scheme in the paper does not support your argument convincingly.
> >
> > > (2) The performance in domain adaptation is calculated after adaptation. That is, after a pretrained model adapts on a target domain, its performance on novel samples from the target domain is calculated. In Test-Time Adaptation, the performance of an adaptation method is calculated in an online manner where the model needs to provide the prediction of a given batch before receiving the next batch from the stream (please refer to the preliminary paragraph in our method section where we formalize the interaction between a TTA method and the stream of unlabeled data)
> >
> > As above, this is not a limiting factor, this is with respect to the evaluation, not with respect to the model. You could compare to an UDA approach and evaluate the performance in the same way. Your claim here seems to be that such a method will perform worse than DISTA, but this is not obvious and should be benchmarked.
> >
> > Overall, I stand with my first assessment that besides an in-depth discussion of related UDA work, empirical evaluation against UDA methods is needed, e.g. in form of another table. If the argument should hold that the evaluation setup is not applicable to UDA, this still needs to be backed up with empirical evidence in the next iteration of the paper.

---

> ### Author Response · Authors · 2023-11-16
> **Official Comment by Authors (Part 3)**
>
> **Regarding the stochasticity of DISTA**: We have repeated the experiments with DISTA (episodic evaluation on Gaussian Noise in ImageNet-C) 10 times, where we found that the maximum difference between two runs (in terms of error rate) is 0.1%. This shows that our method provides consistent results with very negligible stochasticity. We note that this stochasticity is due to either different order of data revealed from the stream (as pointed out in our second distinction between UDA and TTA) and the random sampling from $\mathcal D_s$.
>
> **Regarding including experiments on different models**: We would like to thank the reviewer for this comment. In our experiments, we included 4 different architectures (ResNet50, ResNet18, ResNet50-GN, ViT) which exceeds the experimental complexity of several published works (e.g. [A, B, C]). Further, we considered a total of 8 different TTA baselines.
>
> **Regarding rerunning baselines or copying numbers from original papers**: We re-evaluated the previous method and reproduced their reported results in the original paper. As mentioned in Appendix B.1, we used the original publicly available code for each method.
>
> **Regarding the effect of $\epsilon$ and $E_0$**: We appreciate the reviewer’s comment. We note here that  Both $E_0$ and $\epsilon$ are set following EATA with their recommended hyperparameters. Further,  we conducted  experiments with two values of E_0 and report the results in the table below.
> | E0   | Gauss     	| Shot      	| Impul     	| Defoc     	| Glass     	| Motion    	| Zoom      	| Snow      	| Frost     	| Fog 	| Bright| Contr | Elastic   	| Pixel     	| Jpeg      	| Avg.      	|
> |----------|---------------|---------------|---------------|---------------|---------------|---------------|---------------|---------------|---------------|---------------|---------------|---------------|---------------|---------------|---------------|---------------|
> | 0.4 log(1000)   | 62.9      	| 60.7      	| 61.4 | 65.8      	| 65.9      	| 51.2      	| 46.5      	| 47.6      	| 53.7      	| 39.3      	| 31.7      	| 54.3      	| 41.6      	| 38.5      	| 44.1      	| 51.0      	|
> | 0.5 log(1000)	| 62.2      	| 59.9      	| 60.6      	| 65.3      	| 65.3      	| 50.4      	| 46.2      	| 46.6      	| 53.1      	| 38.7      	| 31.7      	| 53.2      	| 40.8      	| 38.1      	| 43.5	| 50.4
>
> We observe that varying E_0 results in performance variation due to filtering more/less samples from the update step. This result is in Table 11 in the Appendix.
>
> **Regarding ‘limited data’ in our abstract**: We thank the reviewer for this comment. We would like to clarify here that we meant by limited data is the amount of data the stream reveals at a given time step $t$. TTA methods can only leverage this amount of data (which is limited by construction) for adaptation (our first distinction between TTA and UDA). We agree that a wealth of data exists, but it is made available in a sequential manner in test-time adaptation.
>
> **Regarding "conducting several adaptation steps ... will lead to overfitting"**: We appreciate this comment. We note here that since the stream $\mathcal S$ reveals data in batches for the adaptation method, conducting several adaptation steps will make the model overfit to the given batch as all the adaptation steps are performed on the same batch. To confirm this claim, we conducted an experiment with the previous state-of-the-art method EATA where we allow the model to adapt with 1, 2, 3, 4, 5 steps for each received batch from the stream. Results are reported in the table below:
>
> | Number of steps  | Gauss | Shot | Impul | Defoc | Glass | Motion | Zoom | Snow | Frost | Fog  | Bright | Contr | Elastic | Pixel | Jpeg | Avg. |
> |--------|-------|------|-------|-------|-------|--------|------|------|-------|------|--------|-------|---------|-------|------|--------|
> | EATA-1 | 64.0  | 62.1 | 62.5  | 66.9  | 66.9  | 52.5   | 47.4 | 48.2 | 54.2  | 40.2 | 32.2   | 54.6  | 42.2	| 39.2  | 44.7 | 51.9   |
> | EATA-2 | 68.3  | 63.8 | 65.9  | 72.6  | 72.4  | 53.7   | 48.6 | 49.3 | 55.7  | 40.5 | 32.9   | 58.2  | 42.9	| 39.8  | 45.7 | 54.0   |
> | EATA-3 | 74.1  | 70.5 | 75.6  | 86.9  | 81.2  | 58.9   | 52.0 | 51.8 | 60.2  | 41.7 | 34.2   | 74.4  | 45.4	| 41.5  | 48.0 | 59.8   |
> | EATA-4 | 90.4  | 82.1 | 85.7  | 96.8  | 91.5  | 67.1   | 52.9 | 55.5 | 67.2  | 43.0 | 35.2   | 95.7  | 46.1	| 42.0  | 49.0 | 66.7   |
> | EATA-5 | 95.3  | 92.9 | 93.2  | 97.1  | 96.2  | 70.6   | 56.6 | 56.9 | 74.7  | 45.1 | 35.4   | 97.3  | 47.9	| 44.4  | 51.2 | 70.3   |
>
> We observe that the more adaptation steps EATA conducts on each revealed batch, the worse its performance becomes. This experiment confirms our claims and demonstrates the effectiveness of our auxiliary task in improving the performance with the additional adaptation step.
>
> We hope that our answers addressed the reviewers weaknesses and questions. We are happy to engage in further discussion to clarify any confusion or misunderstanding.

---

> ### Author Response · Authors · 2023-11-22
> **Last Day of Discussion**
>
> Dear Reviewer
>
> As the discussion period is about to end (22nd of November), we ask the reviewer to take a look at our response and modifications to the paper to address their questions and weaknesses. We are happy to engage further in discussions to clarify any confusion or misunderstanding.
>
> Best,

---

### Meta-Review · Area_Chair_Jkao · 2023-12-04

**Metareview:**

The paper proposes a new method for test time adaptation, suggesting storing (and using) unlabelled data from the test distribution. I concur with the reviewers that this setup needs to be clarified far better. It is unclear why storing the labels is a problem, especially as we can get extremely high-quality pseudo-labels for them using the model trained on that same data with supervision. The relation with unsupervised domain adaptation is not sufficiently tested empirically, given the similarity between the two settings.

**Justification For Why Not Higher Score:**

The setting is unclear and seems unrealistic (lacks a compelling example, even aspirational). Methodologically, it places the method in between test-time and unsupervised adaptation without comparing or clarifying the similarity.

**Justification For Why Not Lower Score:**

N/A

---

### Decision · Program_Chairs · 2024-01-16

Reject